# Simulation performance of planetary boundary layer schemes in WRFV4.3.1 for near-surface wind over the western Sichuan Basin: a single site assessment

5    Qin Wang[1], Bo Zeng[2], Gong Chen[2] ,Yaoting Li[1]

[1]Civil Aviation Flight University of China, Guanghan, China

[2]Institute of Plateau Meteorology, CMA, Chengdu/Heavy Rain and Drought-Flood Disasters in Plateau and Basin Key Laboratory of Sichuan Province, Chengdu, China

*Correspondence to:* Bo Zeng (bozeng126@126.com)

10    **Abstract.** The topography of Sichuan Basin is complex, high-resolution wind field simulation over this region is of great significance for meteorology, air quality, and wind energy utilization. In this study, the Weather Research and Forecasting (WRF) model was used to investigate the performance of different planetary boundary layer (PBL) parameterization schemes on simulating near-surface wind fields over Sichuan Basin at a spatial resolution of 0.33km. The experiment is based on multiple case studies, so 28 near-surface wind events from 2021 to 2022 were selected, and a total of 112 sensitivity simulations were carried out by employing four commonly used PBL schemes: YSU, MYJ, MYNN2, and QNSE, and compared to observations. The results demonstrate that the wind direction can be reasonably reproduced, its sensitivity to the PBL scheme appears to be less pronounced compared to the near-surface wind speed, though some variability is still observed. As for wind speed, the QNSE scheme had the best performance in reproducing the temporal variation out of the four schemes, while the MYJ scheme had the smallest model bias. Further cluster analysis demonstrates that the sensitivity of the PBL schemes is affected by diurnal variation and different circulation genesis. For instance, when the near-surface wind event caused by the southward movement of strong cold air and occurred during 6:00 and 8:00 (UTC), the variation and speed can be well reproduced by all four PBL schemes and the differences between them are small. However, the simulation results for strong winds occurring during the mid-night to early morning hours exhibit poor root mean square errors but high correlation coefficients, whereas for strong wind processes happening in the early to late evening hours and for southwesterly wind processes demonstrate the opposite pattern. Overall, the four schemes are better for near-surface wind simulations in daytime than at night. The results show the role of PBL schemes in wind field simulation under unstable weather conditions, and provide a valuable reference for further research in the study area and surrounding areas.

## 1 Introduction

Wind, as one of the fundamental natural phenomenon in the atmosphere, poses not only hazards to civil aviation safety and maritime transportation during severe

wind events (Manasseh and Middleton,1999; Leung et al.,2022), but also impacts the dispersion of atmospheric pollutants directly near the surface, leading to adverse effects on public health and the environment (Liu et al., 2020; Coccia, 2020; Yang and Shao, 2021). What's more, wind energy has attracted increasing attention because of its non-polluting and renewable nature, but due to the random nature of wind speed, wind power generation is intermittent, which poses security and stability challenges for large-scale integration of wind energy into the power network( Liu et al., 2019; Kibona, 2020; Shi et al.,2021). Therefore, the accurate prediction of near-surface winds has become the key to ensure traffic safety, optimize wind energy utilization and evaluate air quality, and it is also an important scientific issue for disaster prevention and mitigation, economic benefits and human life and health.

Near-surface wind fields are influenced by a combination of various factors (Zhang et al., 2021), including atmospheric dynamic and thermodynamic processes (such as pressure gradient force, temperature gradients, and so on), topography (such as geographical features, elevation), and underlying surface (such as vegetation, land use). As a state-of-the-art mesoscale weather prediction model, the Weather Research Forecast (WRF) model can predict the fine-scale structure of near-surface wind fields by simulating the evolution of various physical processes in the atmosphere, which is significantly better than the prediction model based on statistics which lacking the description of thermodynamic processes. Furthermore, there are so many researches on the prediction and simulation of the refined characteristics of local wind field by using WRF model (Prieto-Herráez et al., 2020; Salfate et al., 2020; Xu et al., 2020; Tiesi et al., 2021; Wu et al., 2022; Yan et al., 2022; Mi et al., 2023). Although the simulation of near-surface wind fields involves the nonlinear interactions of various physical processes, the physical processes in the planetary boundary layer (PBL) play a direct role in influencing near-surface wind fields. As the interaction area between the atmosphere and the ground, the thermal and dynamic structure, the turbulent motion and mixing process in the boundary layer will directly affect the distribution of the near-surface wind field, so the simulation of the boundary layer by the model can directly affect the accuracy of the near-surface wind field(Chen et al., 2020).

In the mesoscale model, since the employed grid scales and time steps cannot explicitly represent the spatiotemporal scales which turbulent eddies operate on, the PBL parameterization scheme was used to express the effects of turbulent eddies (Dudhia, 2014). The latest version 4.3.1 of WRF model provides more than 10 kinds of PBL parameterization schemes, the differences among them are mainly due to the different methods of dealing with the turbulence closure problem. In China, Ma et al. (2014) conducted a series of sensitivity simulations on spring strong wind events in Xinjiang Province using the YSU, MYJ, and ACM2 schemes. The results indicated that the YSU scheme exhibited greater downward transport of high-level momentum, attributed to enhanced turbulent mixing effects (Hong et al., 2006). The YSU scheme has also been shown to be the optimal PBL scheme for simulating 10-meter wind speeds in other regions (Cui et al., 2018; Li et al., 2018). However, in coastal areas like Fujian Province (Yang et al., 2014), studies have demonstrated that the MYJ scheme is the best choice for simulating near-surface wind speeds due to its

advancements in calculating turbulent kinetic energy (TKE). The MYJ scheme computes TKE at each level, allowing for a more precise representation of turbulence within the boundary layer, which enhances its ability to model the generation, dissipation, and transport of turbulence (Janjié, 1990; Jaydeep et al., 2024). In the mountainous terrain of Huanghan and Guizhou, ACM2 has demonstrated superior performance in simulating near-surface wind speeds (Zhang and Yin, 2013; Mu et al., 2017). From these studies, it is evident that the performance of a PBL scheme is highly dependent on its ability to accurately represent the key physical processes within the boundary layer across different topographical contexts, leading to significant regional variations in the performance of PBL schemes in WRF.

Sichuan Basin is one of the four major basins in China, it is bordered by the Qinghai-Tibet Plateau to the west, the Daba Mountains to the north, the Wushan Mountains to the east, and the Yunnan-guizhou Plateau to the south. Because of the complex terrain of its surrounding areas, the local atmospheric circulation is also complex and unique(Yu et al., 2020), the weather here is characterized by low wind speed, low sunshine and high humidity throughout the year, therefore it is also one of the four major haze areas in China (Li et al., 2021). Under the unique terrain of the Sichuan Basin, it is difficult to determine whether cold air from mid to high latitudes can bypass the Qinghai-Tibet Plateau and then cross the Qinling Mountains to enter the basin. Besides, the basin effect makes it easier to form an inversion structure close to the surface and stabilizing the atmosphere (Gao et al., 2016; Feng et al., 2023). These factors make it one of the regions with the poorest wind forecasting performance in China(Pan et al., 2021; Xiang et al., 2023). Therefore, wind is not still as wildly studied as temperature and precipitation in Sichuan Basin, and numerous studies hitherto have concentrated on the pollutant dispersion under stable and weak wind conditions here, and less attention paid to unstable or strong wind process.

As is known, the interaction between the surface and atmosphere, as well as the characteristics of turbulent motion over the basin terrain, differ from that over plains and plateau areas (Turnipseed et al., 2004; Rajput et al., 2024). However, there has been no comprehensive evaluation of the performance of PBL schemes in simulating the near-surface wind field over the Sichuan Basin, whether using a single measurement site or multiple regional sites. Thus, combing the spatiotemporal refinement requirements from low-altitude flight safety, this study aims to evaluate the performance of four commonly used PBL schemes in reproducing near-surface wind fields with high spatiotemporal resolution by using the wind data from Guanghan Airport in the western Sichuan Basin. So, a horizontal resolution of 0.3 km was used in the model set-up for research, which is a major challenge in such region, because the spatial resolution is in the range of 0.1-1km, which is often referred as "gray zone" in numerical forecasting (Wyngaard, 2004; Liu et al., 2018; Yu et al., 2022). As suggested by many studies, the spatial resolution in "gray zone", is too finely detailed with regarding to the mesoscale turbulence parameterization scheme, and too coarse for the Large Eddy Simulation (LES) scheme to analyze turbulent eddies (Shin and Hong, 2015; Honnert et al., 2016). So far, the impact of different PBL schemes under the spatial resolution of "gray zone" is still uncertain. Hence, a

total of 28 wind events is simulated with a purpose of getting a reliable evaluation, and the study is based on a case study approach, rather than on continuous simulations. In general, this study not only has important significance for improving the wind field forecast in this region, but also provides a scientific basis for the further improvement and development of PBL scheme.

## 2 Data and Method

### 2.1 Data and experimental design

In this study, the experimental approach is different from what has been used in other studies, where one case or long continuous time is simulated. In this study, a total of 28 historical near-surface wind events was simulated by running WRF-ARW (version 4.3.1). We choose Guanghan Airport as the representative of western Sichuan Basin, and the 28 discontinuous windy days, with a criteria of the maximum wind speed greater than 6 m s$^{-1}$ are simulated. The 6 m s$^{-1}$ wind speed threshold is established based on operational considerations specific to Guanghan Airport, particularly regarding the safe conduct of flight training activities with small and medium-sized aircraft. The simulation domain consists of four two-way nested domains of horizontal resolutions 9 km, 3 km, 1 km and 0.33 km, with 105×105, 103×103, 103×103 and 103×103 grid cells, respectively, and 45 vertical levels up to a pressure level of 50hPa was used in all domain, including 10 layers below 2 km. Figure 1 presents the domain set-up. As can be seen from Fig. 1 (a), the outermost domain (D01) covers the western Sichuan Plateau and the northern Qinlin Mountains. The surrounding mountains are mostly between 1,000 and 3,000 meters above sea level, while the basin is between 250 and 750 meters. Due to the complex topography in the upstream region, the influence of cold air on the Sichuan Basin is variable, and the wind simulation is very difficult. In the western domain 2, the elevation gradually decreases from 2000 to 500 meters, with a topography that is higher in the western and northern parts, and lower in the eastern and southern parts. In the domain 4, the transitional zone from plateau to basin is avoided. This area is located in the northern part of Chengdu Plain, and the simulation center is set at Guanghan Airport (104.32° E, 30.93° N). Additionally, Guanghan Airport is located at the western foothills of the Longquan Mountains, only 10km away.

Given the complex terrain in study region and the high resolution of model design, the input of land surface data is particularly important, and its accuracy will directly affect the simulation of land surface processes and atmospheric boundary layer characteristics (Qi et al., 2021). Therefore, we replaced the terrain data of the 4-layer nested area with 3 s resolution (~90 m) from the southwest region of Shuttle Radar Topography Mission (SRTM3)(Farr et al., 2007).

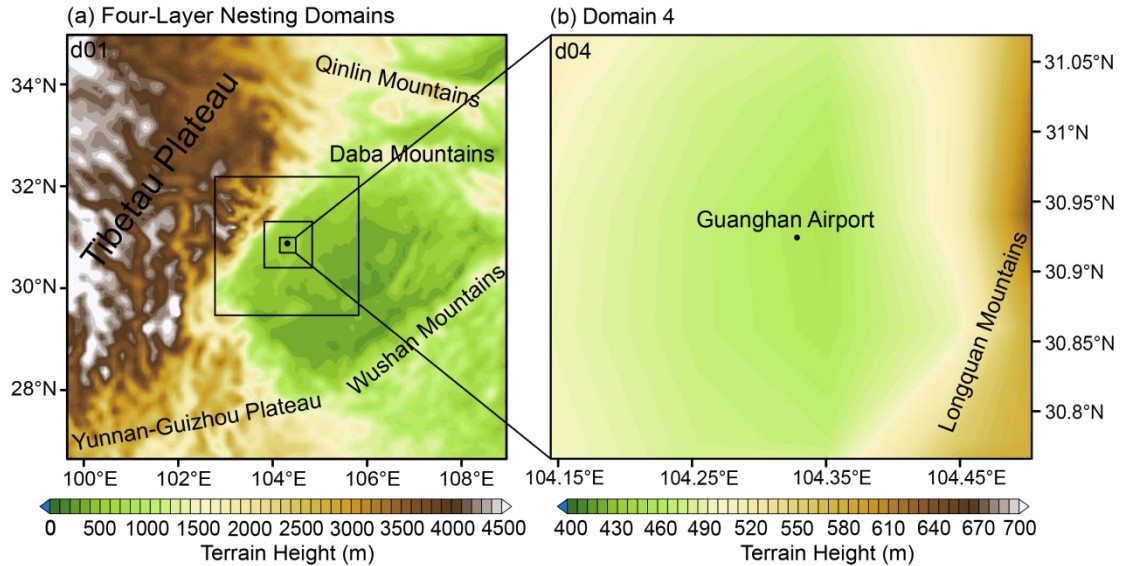

**Figure 1.** Configurations of **(a)** four-layer nesting domains (D01-D04) in WRF and the **(b)** study area. The spatial resolutions are 9, 3, 1 and 0.3 km, for domains D1 to D4, respectively. The figure depicts the actual orography implemented in the experiments.

To evaluate the model's ability in different PBL schemes, the observed wind fields at 10 meters height at Guanghan Airport station is used. The terrain here is flat and homogeneous, and prevailing wind direction are north and northeast in climatology. Wind direction and speed were measured using the FIRST CLASS three-cup anemometer and wind vane, both manufactured by Thies Clima inc. in Germany. The anemometer has a measurement range of 0.3 to 75 m s$^{-1}$ and a starting threshold of less than 0.3 m s$^{-1}$, with an accuracy of 1% of the measured value or less than 0.2 m s$^{-1}$. The wind vane covers a measurement range of 0 to 360°, with a starting threshold of less than 0.5 m s$^{-1}$ at a 10° amplitude (as per ASTM D 5366-96) and 0.2 m s$^{-1}$ at a 90° amplitude (according to VDI 3786 Part 2), and an accuracy of 0.5°. During the research period, the anemometers were annually calibrated by accredited institutions. Before incorporating the wind data into our analysis, we performed basic data checks and quality control procedures, including outlier removal.

The hourly reanalysis datasets ERA5 with a horizontal resolution of 0.25° and 38 vertical levels, is used to provide the initial and boundary conditions for WRF simulations, which are updated every 3 hours when input into the model. Each event is simulated using four different PBL parametrisation schemes. Thus, a total of 112 simulations are carried out. Each simulation spans 24 hours, with the corresponding high winds in the middle of the simulation, and discarding a spin-up period of 3 hours, and the model results are output every 10 minutes, enabling a high temporal resolution for demanding, the other model configuration is summarized in Table 1.

**Table 1.** Configures of the physics scheme in WRF simulation.

| Parameterizations | Configuration |
|---|---|
| Microphysics scheme | WSM 3-class graupel scheme (same for each domain) |
| Longwave radiative scheme | RRTM shortwave (same for each domain) |
| Shortwave radiative scheme | Dudhia shortwave (same for each domain) |
| Cumulus convection scheme | Kain-Fritsch for the outermost domain, and closed in other 3-layers |

## 2.2 PBL Schemes

There are more than 10 PBL parameterization schemes in WRF-V4.3.1, but four commonly used PBL schemes were selected for this study, which are YSU (Yonsei University) scheme (Hong et. al., 2006), MYJ (Mellor-Yamada-Janjic) scheme (Janjié, 1990), MYNN2 (Mellor-Yamada- Nakanishi-Niino Level 2) scheme (Nakanishi and Niino , 2009) and QNSE (Quasi-Normal Scale Elimination) scheme (Sukoriansky and Galperin, 2006). Among them, YSU is a non-local, first-order closure scheme that represents entrainment at the top of the PBL explicitly, while the rest are local closure scheme, detail characteristics can be seen in Table 2. The surface layer scheme in the experiment is matched with each PBL scheme.

**Table 2.** The four selected PBL schemes and surface schemes in experiment.

| PBL scheme | Advantages | Surface layer scheme | Land surface scheme |
|---|---|---|---|
| YSU | 1st-order closure scheme that is widely utilized for its robust representation of turbulence closure processes (Hong et. al., 2006). | Revised MM5 Monin -Obukhov scheme | Noah MP |
| MYJ | A 1.5-order closure scheme that is known for its effectiveness in capturing vertical mixing processes (Janjié, 1990). | MYJ | Noah MP |
| MYNN2 | A 1.5-order closure scheme that improves the simulation of sub-grid scale turbulence (Nakanishi and Niino, 2009). | MYNN | Noah MP |
| QNSE | A 1.5-order turbulence closure scheme that accounts for both turbulent and | QNSE | Noah MP |

non-turbulent mixing
processes in the
atmosphere
(Sukoriansky and
Galperin, 2006).

## 2.3 Statistical metrics for validation

As suggested by Wang et al. (2017), different sky conditions and atmospheric stability will affect the simulation of wind fields. So, in order to accurately evaluate the sensitivity of four PBL schemes to the near-surface wind field in the western Sichuan Basin on the east side of the Qinghai Tibet Plateau, 28 near-surface wind cases are selected for simulation based on wind speed data at 10-minute intervals

from 2021 to 2022, when the 10 minutes averaged wind speed greater than or equal to 6 m s$^{-1}$ last for 30 minutes, and the result is evaluated separately through different circulation patterns and K-means clustering analysis method. The main statistical metric used includes:

        Root Mean Square Error (RMSE), which is the square root of the average of the

squared differences between the simulated and observed values. RMSE is a commonly used metric in model evaluation, assigning higher weight to cases with larger simulation errors:

$$\text{RMSE} = \sqrt{\frac{\sum_{i=1}^{N}(O_i - S_i)^2}{N}} \tag{1}$$

        where $N$ is the total number of samples, $O_i$ represents the observed near-surface

wind, and $S_i$ denotes the simulated near-surface wind, measured in m s$^{-1}$.

        Correlation Coefficient (COR) is an indicator that measures the strength and direction of the linear relationship between simulation and observation. By analyzing COR, the consistency between simulation results and observation results can be evaluated, and the corresponding PBL scheme can accurately capture the variation

relationship of ground wind speed:

$$\text{COR} = \frac{\sum_{i=1}^{N}(s_i - \bar{s})(O_i - \bar{O})}{\sqrt{\sum_{i=1}^{N}(s_i - \bar{s})^2 \sum_{i=1}^{N}(O_i - \bar{O})^2}} \tag{2}$$

        where $N$ is the total number of samples, $O_i$ represents the observed values, and $S_i$ denotes the simulated values.

        BIAS refers to the average difference between simulated and observed values,

reflecting the overall bias of the simulation results. If BIAS is close to 0, it indicates that the simulation results have good accuracy at the average level. The calculation formula is as follows:

$$\text{BIAS} = \frac{1}{N}\sum_{i=1}^{N}(S_i - O_i) \tag{3}$$

The Weibull distribution is a probability function used to describe the distribution of wind speed (Lai, et al., 2006; Jiang, et al., 2015). The expression for the Weibull distribution probability density function of wind speed $v$ is:

$$f(v) = \frac{\kappa}{\lambda}\left(\frac{v}{\lambda}\right)^{\kappa-1} exp\left[-\left(\frac{v}{\lambda}\right)^{\kappa}\right] \qquad (4)$$

where $k$ is the shape parameter, a dimensionless parameter, and $\lambda$ is the scale factor, measured in m s$^{-1}$. These two parameters can be calculated using the following formulas:

$$\kappa = \frac{\sigma}{\mu} \qquad (5)$$

$$\lambda = \frac{\mu}{\left(0.568+\frac{0.434}{k}\right)^{\frac{1}{k}}} \qquad (6)$$

where $\sigma$ and $\mu$ represent the standard deviation and mean value of the wind speed, respectively.

## 3. Overview of historical cases and evaluation of simulation results

### 3.1 Summary of 28 near-surface wind events

Since the experiment approach is concerned about multiple cases simulation in this study, it is necessary to understand the characteristics of these cases, such as the temporal variation, the peak time and synoptic factors, which can help to classify them and evaluate their simulation performance separately in the following analysis.

Therefore, Table 3 provides detailed information derived from wind data recorded at 10-minute intervals. It is shown that out of the 28 near-surface wind events participating in the simulation, 24 were northerly events, accounting for 85% of the total. The events in which the maximum wind is above 8 m s$^{-1}$ accounts for 18%, and the events of 5-7 m s$^{-1}$ accounts for 82%. Meanwhile, the wind direction corresponding to the peak time was distributed between 350 ° -50 °, with northeasterly winds between 0-50 ° being the most common. Additionally, the left are 4 southerly winds cases, all of which appear to occur in summer or early autumn.

As for the dominated factors of each event, the term 'cold air' in Table 3 was used to denote the cases which are generated by incursion of cold air from northern regions like Siberia or Mongolia in Sichuan Basin, often accompanied by sharp temperature drop and changes in humidity. The term 'convective system' specifically denotes the strong wind cases primarily caused by convective weather systems, often accompanied by thunderstorm. In such cases, the vertical motion or convection is the dominant. It is shown that most of the wind events were mainly caused by incursion of cold air, only little were associated with convective weather systems. Influenced by this, the spring (March-May) process accounted for the most, accounting for 46%, followed by summer and autumn, both accounting for 25%. In terms of the peak time, 60% of the simulated cases appear to concentrate on 5:00 - 9:00 UTC and 10:00 -

14:00 UTC at night, then followed by 15:00 - 19:00 UTC, and there are a total of 6 events occurred at 20:00 - 23:00 UTC and 0:00 - 4:00 UTC, accounting for 21%.

The near-surface wind speed in the Sichuan Basin exhibits a distinct diurnal variation, characterized by lower wind speeds in the morning and evening and higher wind speeds at midday. In order to analyze the temporal variation of wind speed under different conditions, the hourly time series of the observed wind speed for 28 cases is presented in Fig. 2. It is showed that many cases with the incursion of cold air exhibit diurnal variation characteristics. Because, in these cases, cold air predominantly affects the western Sichuan Basin around midday (Table 3). However, for strong wind events such as cases No. 9, 13, 25, and 26, which were caused by convective systems, there was no clear diurnal variation in wind speed, and is characterized by sudden changes in wind speed, reflecting the transient and localized nature of convective processes.

**Table 3.** Characteristics and circulation patterns of the 28 chosen near-surface wind events.

| Event ID | Date yyyy-mm-dd | Maximum wind speed (m s$^{-1}$) /direction(°) | Maximum wind time hh:mm | Impact Factor |
|---|---|---|---|---|
| 1 | 2021-03-17 | 6.0/350° | 09:40 | Cold air |
| 2 | 2021-03-24 | 6.8/350° | 08:00 | Cold air |
| 3 | 2021-03-30 | 6.1/90° | 09:50 | Cold air |
| 4 | 2021-03-31 | 6.4/45° | 09:00 | Cold air |
| 5 | 2021-04-23 | 6.3/47° | 11:00 | Cold air |
| 6 | 2021-04-25 | 7.0/70° | 08:00 | Cold air |
| 7 | 2021-04-27 | 8.3/18° | 11:10 | Cold air |
| 8 | 2021-06-16 | 6.9/46° | 07:40 | Cold air |
| 9 | 2021-07-21 | 7.1/158° | 06:20 | Convective system |
| 10 | 2021-08-22 | 8.0/47° | 03:10 | Cold air |
| 11 | 2021-08-25 | 6.1/33° | 06:00 | Cold air |
| 12 | 2021-09-15 | 6.6/50° | 15:20 | Cold air |
| 13 | 2021-09-19 | 6.0/183° | 08:00 | Convective system |
| 14 | 2021-09-25 | 6.1/54° | 05:00 | Cold air |
| 15 | 2021-10-01 | 6.0/332° | 14:40 | Cold air |
| 16 | 2021-10-04 | 7.3/45° | 03:30 | Cold air |
| 17 | 2021-11-06 | 9.6/51° | 12:00 | Cold air |
| 18 | 2021-12-25 | 6.0/46° | 20:50 | Cold air |
| 19 | 2022-03-19 | 7.9/10° | 22:10 | Cold air |
| 20 | 2022-03-30 | 8.3/43° | 12:20 | Cold air |
| 21 | 2022-04-14 | 6.0/27° | 18:40 | Cold air |
| 22 | 2022-04-27 | 8.3/50° | 17:00 | Cold air |
| 23 | 2022-05-08 | 7.1/26° | 17:30 | Cold air |
| 24 | 2022-05-13 | 9.2/40° | 22:40 | Cold air |

| 25 | 2022-06-23 | 6.2/119° | 11:10 | Convective system |
| 26 | 2022-08-17 | 8.6/148° | 14:40 | Convective system |
| 27 | 2022-08-28 | 6.7/40° | 13:20 | Cold air |
| 28 | 2022-10-03 | 8.5/43° | 02:40 | Cold air |

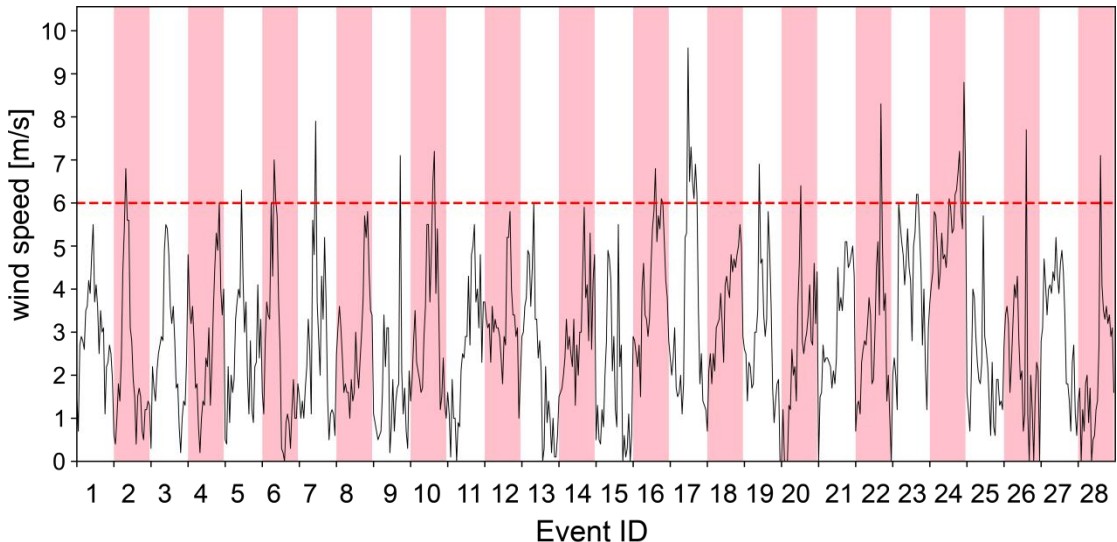

**Figure 2.** The time series of hourly wind speed for all the 28 near-surface wind events
listed in Table 3, each event represents one day, the label of x-axis represents the
event ID shown in Table 3, the shading was employed to distinguish the time series of
the 28 selected cases, which are discontinuous across days.

### 3.2 Overall simulation performance of 28 wind events

First, the performance of the model in different PBL schemes is assessed with
respect to wind direction. Thereby, the simulated wind rose of four PBL schemes are
given in Fig. 3. By comparing with the observation (Fig. 2), it is found that four PBL
schemes can reproduce the distribution of wind direction. Specifically, the simulated
wind directions are basically distributed in NNW, N, NNE, NE and ENE, reproducing
the characteristics of highly concentrating on NNE and NE. Besides, it is also shown
that the occurrence frequencies of the wind fields simulated by all PBL schemes in
the NNE and NE directions are all relatively higher than observation, but for wind in
NNW direction, the simulated frequencies are significantly lower, indicating an
clockwise bias which may be related to the plateau topography with steep terrain in
the northwest and west. The statistical metrics (Gómez et al., 2015)
in simulated 10-m wind direction are also given in Table 4. From the perspective of
BIAS, RMSE, and Circular COR, the differences in wind directions among the four
PBL schemes are relatively small. However, these differences are not negligible and
suggest that while the impact of different PBL schemes on wind direction is minor, it
is observable. Therefore, it can be inferred that the wind direction of the near-surface
wind field in western Sichuan Basin shows some sensitivity to the selected PBL
scheme, though the variations are moderate.

However, there are still some differences in wind direction simulations among four PBL schemes. In MYJ scheme, the frequency of NNE wind is higher than NE wind, which is consistent with the observations. Moreover, the frequencies of N wind and NE wind are closer to the observations. Therefore, MYJ has the best simulation of wind direction. The wind direction distribution simulated by the MYNN2 scheme is very close to QNSE scheme , but due to the worse performance in simulating NNW wind and the larger frequency of simulated NNE and NE wind, MYNN2 scheme is the worst among the four schemes. In general, for wind fields with weather processes passing through, more attention is paid to the simulation of wind speed. So, we will focus on the performance of wind speed next.

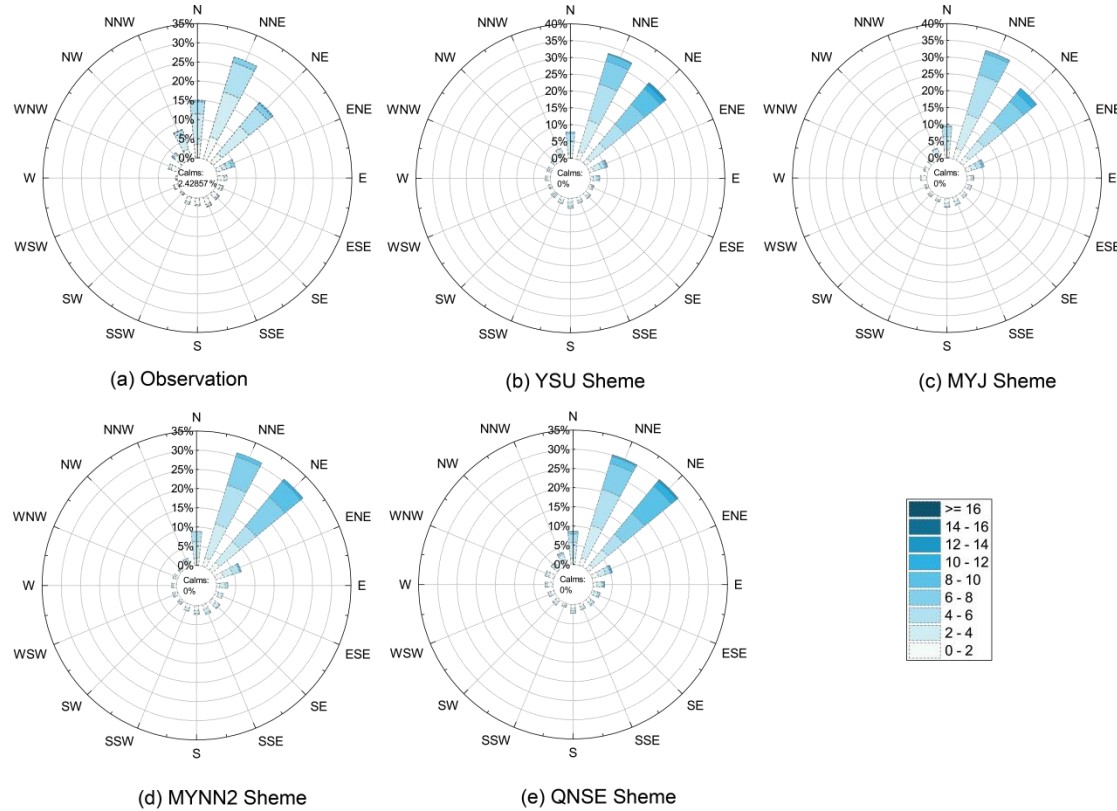

**Figure 3.** The wind rose chart for all the observed and simulated 28 near-surface wind events listed in Table 3, (a) for observation, (b) for YSU scheme, (c) for MYJ scheme, (d) for MYNN2 scheme, and (e) for QNSE scheme, the circles represent the relative frequency (%), and the colors represent wind speed.

**Table 4.** Statistical metrics for simulated 10-m wind direction.

| | Average Wind Direction (°) | BIAS(°) | RMSE(°) | Circular COR |
|---|---|---|---|---|
| Observations | 22.2 | | | |
| YSU | 33.3 | 12.1 | 57.8 | 0.37 |
| MYJ | 32.1 | 12.5 | 58.9 | 0.36 |

| MYNN2 | 36.9 | 14.2 | 61.3 | 0.33 |
| QNSE | 31.0 | 9.8 | 62.1 | 0.30 |

In fact, by comparing Fig. 2 and Fig. 3, it seems that all the four PBL schemes exhibit obvious exaggeration of wind speed, which is also shown in other numerous studies (Dzebre et al.,2020; Ma et al., 2024). For instance, in the research by Yu et al. (2022), all 11 WRF PBL schemes overestimate near-surface wind speeds by approximately 1 m s$^{-1}$ in the Hebei Plain. Similarly, in the experiment conducted by Gómez et al. (2015), the MYJ scheme strongly overestimates the maximum wind speed by more than 10 m s$^{-1}$ at 50% of the locations, while the YSU scheme shows deviations greater than 3 m s$^{-1}$. But, what are the specific simulation characteristics of these commonly used PBL schemes in the Sichuan Basin? To further evaluate the advantages and disadvantages of each scheme in simulating near-surface wind speed, three statistical metrics (COR, RMSE and BIAS) were calculated. These statistics were derived from data recorded at 10-minute intervals across 28 distinct events, as illustrated in Fig. 4. In terms of COR, both the mean and median values for all schemes fall within the range of 0.4 to 0.6, which indicates a tendency for the COR to cluster around this range across the events.Moreover, the median is above the mean value, indicating that the correlation coefficients are all negatively skewed distribution, that is, the correlation coefficients between simulated and observed wind speed are higher than the mean value in most cases, but very poor in some certain cases. It is further illustrated by the heat map displayed in Fig. 4d, where cases No. 3, 11 and 20 demonstrate correlation coefficients below 0. In contrast, QNSE shows the best mean correlation coefficient of 0.6, suggesting the best performance in reproducing the temporal variation of observed wind speed in most cases.

Although there is little difference between the simulated and the observed wind speed in the RMSE and BIAS, it is noteworthy that MYJ scheme has the smallest mean RMSE and BIAS (2.3 and 1.2 m s$^{-1}$) while QNSE has the largest (2.7 and 1.8 m s$^{-1}$). The BIAS is consistent with RMSE as illustrated in the Fig. 4 (c), except that the median and mean BIAS is not as close as RMSE shows in MYJ scheme, indicating that the systematic error (BIAS) might be either too high or too low in certain cases. However, overall, MYJ scheme is highly precise and has little variance in its performance, which is crucial for accurate weather forecasts. The main reason for this may be associated with the basin topography, because the boundary layer is in stable condition in most time, the turbulence is mainly generated and maintained by wind shear, so that the situation showing strong locality. Therefore, the simulation error obtained by MYJ scheme is the smallest in this stable and weakly stable boundary layer, which is consistent with the research results of Zhang et al. (2012). Besides, the result that QNSE scheme has the best performance on capturing the temporal variation of wind speed, maybe because that QNSE scheme improves simulation of sub-grid scale turbulence, and considers more complex and detailed physical processes. Under stable atmospheric stratification, QNSE adopted *k-ε* model developed from turbulent spectral closure model, while under the unstable situation,

the method of MYJ scheme is used, so QNSE scheme has more advantages in the simulation of wind speed variation trend. However, the specific causes require further investigation in future works.

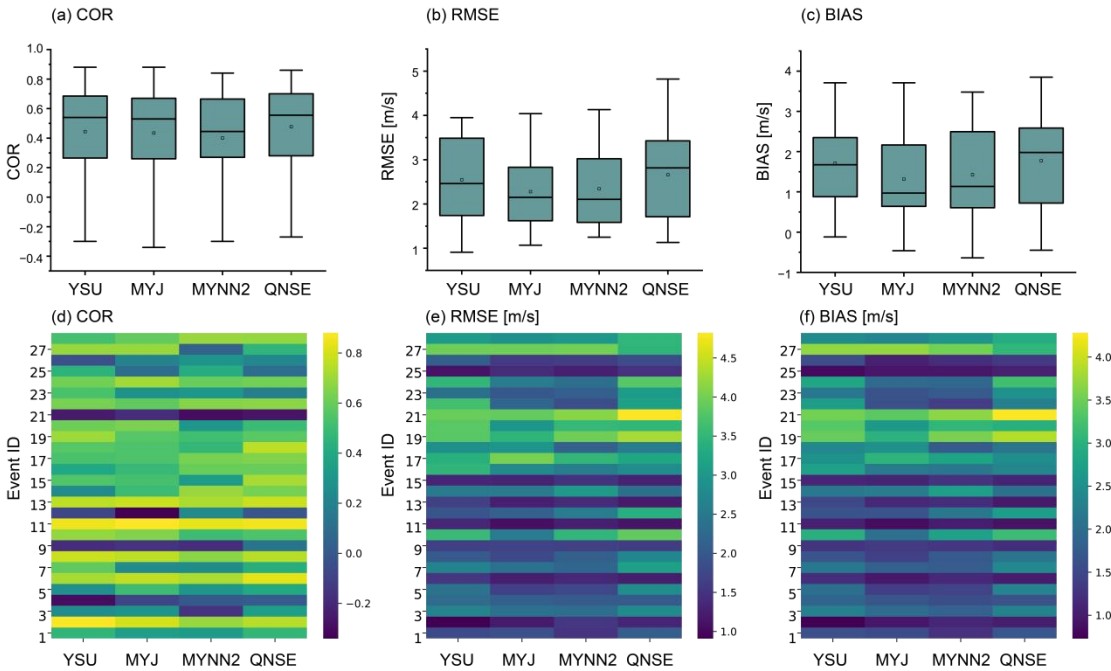

**Figure 4.** Different performance metrics for the comparison of observed and simulated near-surface wind speed at 10-minute intervals for 28 events. Box plots shows the overall characteristics of COR, RMSE and BIAS, and heat-map gives details for certain case. The box represents the metrics range from first quartile to third quartile ,and the line inside the box represents the median, while the empty 385 square represents the mean.

## 3.3 Differences of wind velocity segments and diurnal variations simulated by four PBL schemes

Figure 5 shows the frequency distribution of different winds with the observed and the simulated wind data at Guanghan Airport. As can be seen, the observed wind 390 speed distribution is left-skewed, primarily due to the concentration of wind speeds within the 1-4 m s$^{-1}$ range, where the cumulative frequency exceeds 0.6. When comparing the spread of each PBL scheme's distribution to the observations, all four PBL schemes exhibit a wider distribution, indicating overestimation of the wind speed variability.

In order to quantitatively compare the performance of the four PBL schemes, Weibull distribution fitting was applied in Fig.5. The shape parameter ($k$) represents the concentration of the wind speed distribution. A lower $k$ value indicates a more dispersed distribution with greater wind speed variability, while a higher $k$ value suggests a more concentrated distribution with less variability. The observed $k$ value 400 is 1.79, while the shape parameters for YSU, MYJ, MYNN2, and QNSE are 1.89, 1.83, 1.93, and 1.77, respectively. With a shape parameter of 1.77, QNSE is closest to

the observed value, indicating it captures the variability of wind speeds more effectively than others. Therefore, from the shape parameter perspective, QNSE provides the most similar wind speed distribution to the observations. Conversely, YSU and MYNN2 yield higher $k$ values, suggesting a more concentrated distribution that underestimates variability.

The scale parameter $\lambda$, representing the spread of wind speeds, shows systematic overestimation for all PBL schemes. The observed $\lambda$ is 3.30 m s$^{-1}$, while the scale parameters for YSU, MYJ, MYNN2, and QNSE are 5.20 m s$^{-1}$, 4.69 m s$^{-1}$, 4.88 m s$^{-1}$, and 5.25 m s$^{-1}$, respectively. So, MYJ and MYNN2 exhibit smaller deviations in $\lambda$, indicating closer alignment with observed wind speeds, whereas YSU and QNSE show the largest overestimation.

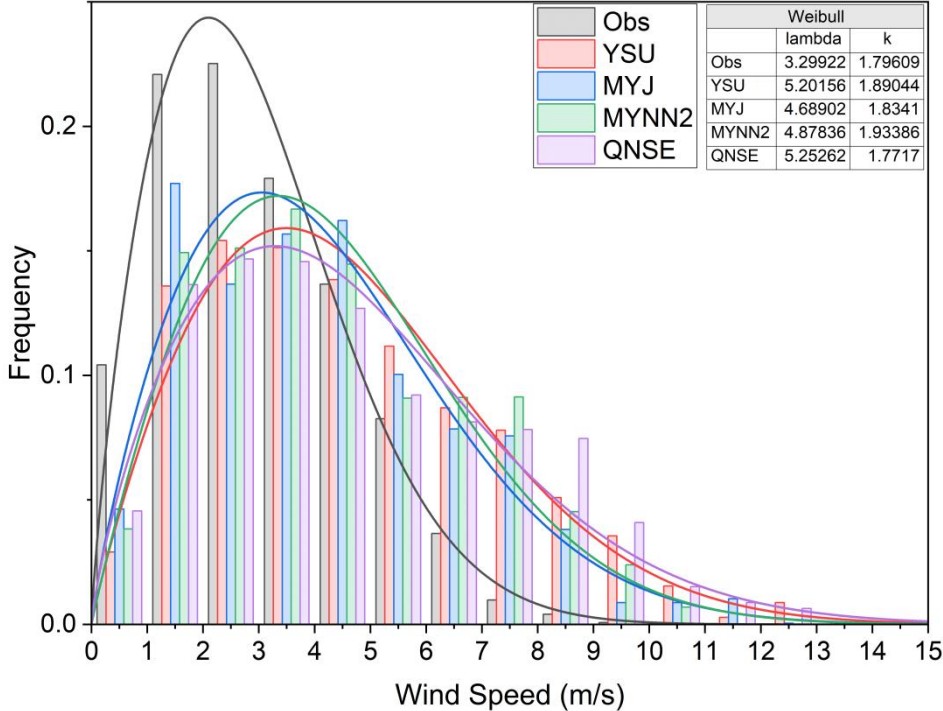

**Figure 5.** The frequency distribution of different wind speeds and Weibull fitting curves for the observed and simulated wind speeds from four PBL schemes, sampled every 10 minutes during 28 wind events. The shape parameter is denoted by ( $k$ ), and the scale parameter by ( lambda ). Each colored line and bar represents one of the PBL schemes.

The performance of the PBL schemes varies across different wind speed ranges. When wind speed below 3 m s$^{-1}$, none of the PBL scheme has a good performance, and the lower the wind speed, the greater the bias. In the wind speed range of 3-5 m s$^{-1}$, different PBL schemes show significant differences compared with observations. Specifically for wind speeds between 3-4 m s$^{-1}$, the simulation results of the MYJ scheme are closest to the observations, followed by MYNN2. For wind speeds between 4-5 m s$^{-1}$, YSU and MYJ simulations are closer to the observations, indicating better performance in this wind speed range. All schemes tend to overestimate when wind speed above 5 m s$^{-1}$. Figure 6 further provides the deviations

between the observed and simulated wind speed of four PBL schemes in different
wind speed ranges. As can be seen, the performance of four PBL schemes differ
greatly with the increase of wind speed, and the wind speed deviation of the same
PBL scheme also increases. For the wind speed below 3 m s$^{-1}$, the simulated wind of
each PBL scheme are about 1.5-2 m s$^{-1}$ higher than the observation. In terms of mean
values, the MYJ scheme exhibits relatively smaller deviations for wind speeds below
435 8 m s$^{-1}$, an average deviation ranging from 0.5 to 1.25 m s$^{-1}$. In contrast, for wind
speeds above 8 m s$^{-1}$, the MYNN2 scheme demonstrates the smallest deviation, with
an average deviation of 2 m s$^{-1}$.

In general, the fitting curve of the QNSE scheme is closest to the observation,
and the $\lambda$ value is slightly to the right of the mode. However, it is critical to highlight
that the MYJ scheme matches observations better than the other schemes in terms of
wind speed. As shown in both Fig. 5 and 6, the MYJ scheme consistently exhibits a
lower error across various wind speed ranges and aligns more closely with the
observed frequency distribution. While all schemes show modes to the right of the
observed wind speed distribution, the MYJ scheme demonstrates a performance that
is closest to the observed data, indicating a tendency towards a more accurate
representation of wind speeds.

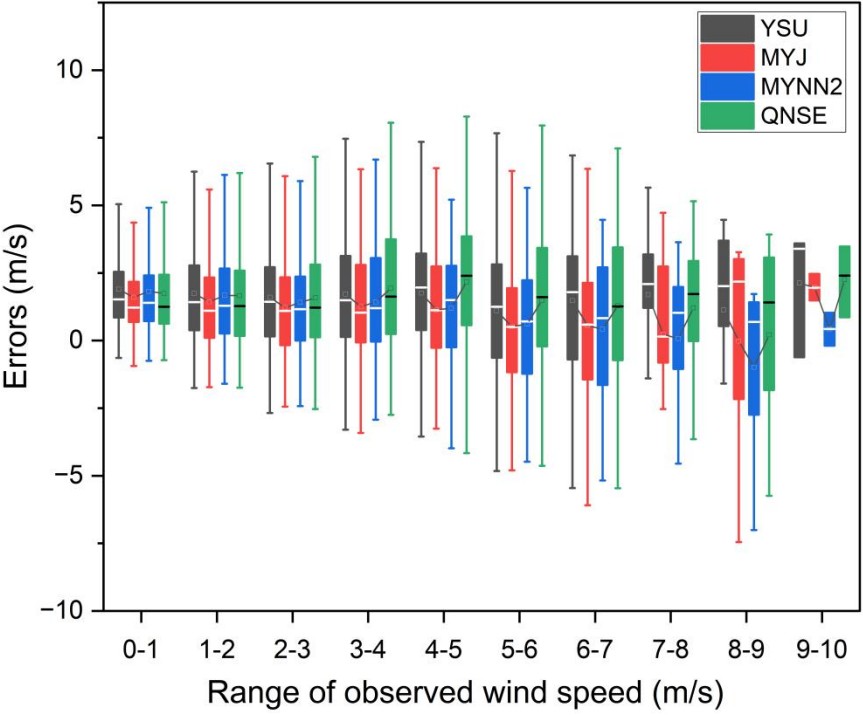

**Figure 6.** Wind speed errors of four PBL schemes in different wind speed segments
for 28 wind events with 10-minute intervals,the line inside the box represents the
450 median.

The variation of near-surface wind field is easily affected by surface
characteristics, especially ground heating. When the weather background is fixed, the
change of local thermal characteristics in a day will inevitably affect the near-surface
wind field. Therefore, there will be significant differences in the wind field simulation

during different time periods between different PBL schemes. In this study, since the study area is located in a time zone of UTC +8 hours (local time), the 'daytime' and 'nighttime' periods were defined in terms of Coordinated Universal Time (UTC). The daytime corresponds to 00:00 - 10:00 UTC, and the nighttime refers to 11:00 to 23:00 UTC. Figure 7 presents the diurnal variation characteristics of wind speed
deviations simulated by the four PBL schemes in the WRF model through box plots.

   In terms of the mean, the performance of each scheme in simulating wind speed tends to be better during the daytime than at night. During the daytime, the MYJ schemes perform relatively well, particularly around local noon at 4:00 UTC, where the errors are lowest (0.76 m s$^{-1}$), indicating that this scheme provide more stable and
reliable simulations during this period. The highest deviations are observed at 18:00 UTC (with errors peaking at 2.80 m s$^{-1}$, followed by 19:00 and 20:00 UTC (2.62-2.63 m s$^{-1}$ ) , indicating that the strong winds occurring during these times are not well simulated by any of the schemes. For the YSU scheme (gray color), the simulation capability is best around local noon (4:00 UTC), indicated by relatively lower mean
errors of 1.02 m s$^{-1}$. This suggests that the YSU scheme effectively captures wind speed closer to local noon. The MYJ scheme (red color) shows reliable performance both at noon and in the evening, with errors 0.75-1 m s$^{-1}$, indicating robust simulation during these periods. The MYNN2 scheme (blue color) performs similarly well in the evening, with the lowest mean errors of 0.66 m s$^{-1}$ at 12:00 UTC. The QNSE scheme
(green color), although showing little variation during the daytime, also demonstrates its best performance at noon with minimal mean errors of 1.18 m s$^{-1}$ (4:00 UTC). This consistent daytime performance suggests reliable outputs for various strong wind events during this period. However, the QNSE scheme exhibits increased variability during nighttime simulations, with errors varying more significantly.

In summary, the performance of the PBL schemes varies based on the time of day, hinting that they may be sensitive to diurnal changes in atmospheric conditions, each PBL scheme displays distinct performance characteristics, with MYJ scheme showing particularly consistent and reliable performance during the daytime, especially around local noon (4:00 UTC), where the mean error is minimized.

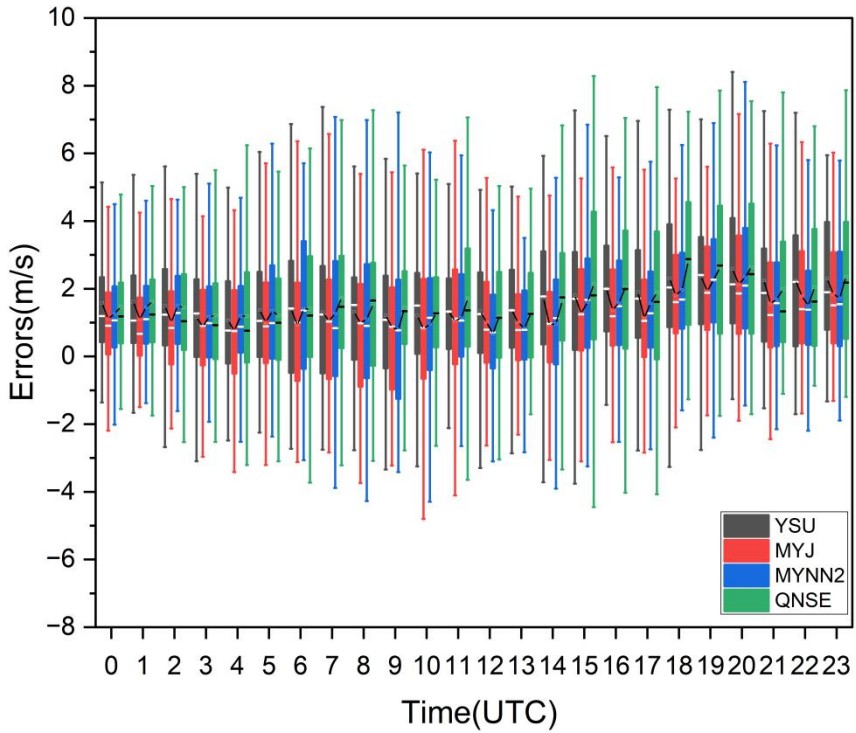

**Figure 7.** Diurnal variation of wind speed errors corresponding to four PBL schemes. The line inside the box represents the mean, while the black short line connects the mean values of each PBL scheme at each hour. Statistics are derived from the data at 10-minute intervals.

**3.4 K-means clustering analysis and performance in different types of events**

From the previous analysis, it is known that as the horizontal grid spacing of 0.33 km is within the PBL gray zone resolution, QNSE scheme can better capture the trend of near-surface wind events over western Sichuan Basin, while the bias produced by MYJ scheme is the minimum. The results also show differences across various wind speed ranges and different time periods in this region. Given the complexity of meteorological conditions in this area, the performance of different PBL schemes may vary under different circumstances. However, directly classifying cases based on weather conditions has not yielded clear insights, partly due to the large differences in sample sizes across categories(Table 3). Therefore, to more effectively evaluate the specific performance of the PBL schemes in simulating near-surface wind events, it is necessary to further classify the 28 cases based on model performance metrics, which can provide a more reliable and meaningful distinction of the schemes' capabilities.

The K-means cluster method based on the RMSE and COR of the four PBL schemes is used to divide the simulation results of 28 near-surface wind events into three categories, as presented in Fig. 8. The RMSE of the cluster center of the first class is 1.9 m s$^{-1}$, and the COR is 0.2. A total of 10 events belong to this class, characterized by moderate RMSE but poor COR. For the second class, the cluster center has an RMSE of 2.85 m s$^{-1}$ and a COR of 0.6. This class includes 12

events, indicating higher RMSE but better COR. The remaining 6 events fall into the third category, where both RMSE and COR are optimal for simulation. The cluster center for this class has an RMSE of 1.25m s$^{-1}$ and a COR of 0.76.

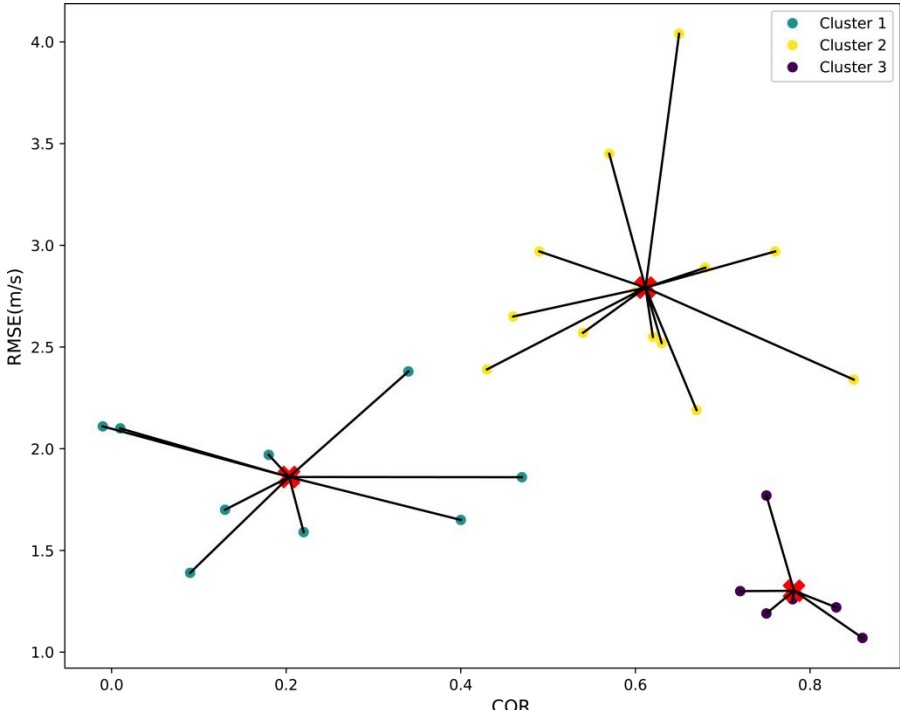

**Figure 8.** Scatter plot of K-means cluster analysis, the red cross symbol represents the cluster center.

Furthermore, it is shown that among these three categories, the QNSE scheme has the best simulation correlation coefficient, while the MYJ scheme has the smallest wind speed simulation error. This consistency in performance is in line with the results prior to applying K-means clustering, indicating that QNSE and MYJ schemes are relatively robust and reliable choices for the near-surface wind simulation in western Sichuan Basin with model grid resolution of 0.3 km. Detailed information on the individual cases corresponding to each cluster can be found in Fig. 9.

Figure 9 shows the RMSE and COR heat-maps of three types of events after cluster analysis, and peak time of gale is specially marked, it is found that different PBL schemes are very sensitive to the diurnal variation. The events in class III is characterized by that the gale period basically occurs between 6:00 and 8:00 UTC, a period known for the maximum surface temperatures and the most unstable atmospheric stratification during the day. This period is characterized by strong surface heating that drives convective turbulence, which leads to vertical mixing and relatively strong near-surface winds. This dynamic makes it easier for models to capture wind profiles accurately. What's more, in the events of class III, except for one thunderstorm gale event, the rest are all typical strong cold air induced near-surface wind processes, which indicates that the four PBL schemes have the good performance in simulating the typical strong cold air wind event occurred in the afternoon. As shown in Fig. 10, the RMSE ranges from 0.21m s$^{-1}$ to 0.96 m s$^{-1}$, and

the COR ranges from 0.05 to 0.19, with only one case having a difference of 0.3, which means that there is little difference between four PBL schemes.

**Figure 9.** Heat-map about the RMSE (numbers) and COR (coloring) of four PBL schemes for 28 near-surface wind simulations according to the cluster analysis. The information in the right column is gale moment (numbers) and classification label (coloring).

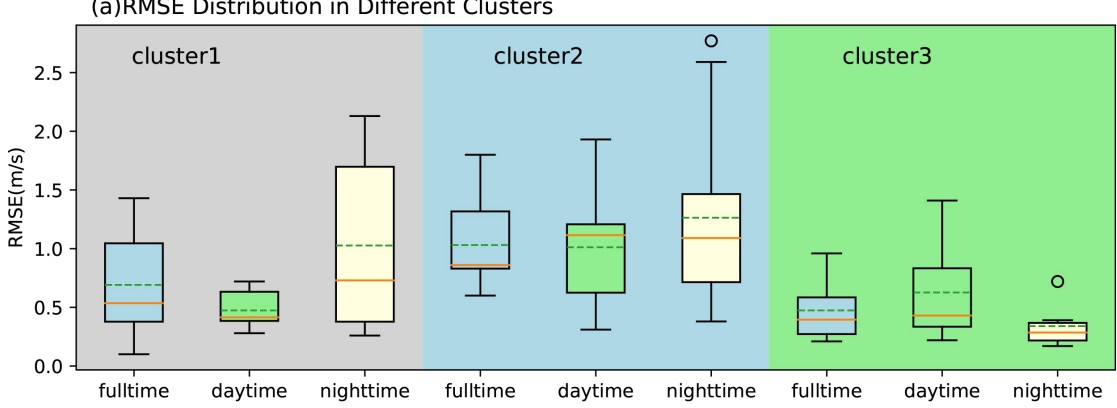

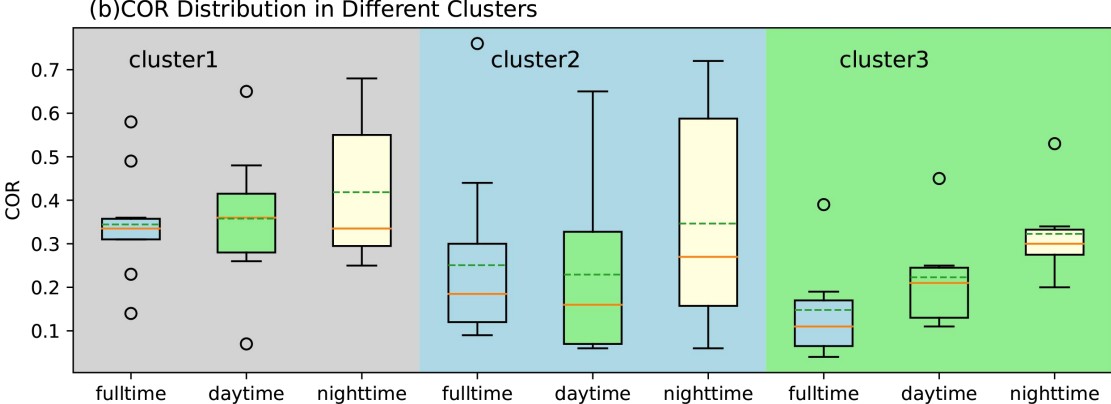

**Figure 10.** Box plots of the maximum differences during four PBL schemes in three types of events, with the green dotted line as the mean, the orange solid line as the median, and the circle as the outlier.

The most obvious differences among the four PBL schemes are mainly in the events of class I and II. Except for one southerly gale event belonging to class III, the other southerly wind events are classified into class I, indicating that the four PBL schemes often have good RMSE and poor COR for southerly wind events caused by convection in western Sichuan Basin. In Fig. 9, it is shown that for class I, the maximum wind speeds most frequently occur during the two periods of 10:00 - 11:00 UTC and 15:00 - 16:00 UTC, with only two cases occurring between 6:00 - 7:00 UTC. The period from 10:00 to 16:00 UTC corresponds to the transition of the atmospheric stratification in the basin from unstable to stable conditions, during which the inversion layer is established. For these events, the difference between the maximum and minimum RMSE and COR obtained by different PBL schemes is as large as 1.43 m/s and 0.58, respectively.

The simulation events of class II exhibit the most significant differences among the four PBL schemes, with characteristics such as gale occurrence times differing markedly from those in class I and class III. It is observed that the four PBL schemes often display high correlation coefficients (COR) and high RMSE for near-surface wind events occurring in the early morning (17:00-22:00 UTC) and early afternoon (3:00-5:00 UTC). Especially in the early morning, the boundary layer typically

experiences a stable stratification due to radiative cooling, which suppresses vertical mixing, and near-surface winds weaken significantly, leads to the highest RMSE due to the models' inability to accurately simulate the disturbances and small-scale dynamics in this stable period. In this type of event, the maximum differences in RMSE and COR among the PBL schemes can reach 1.49 m s$^{-1}$ and 0.76, respectively. In addition, Fig. 10 shows that the differences between different PBL schemes in class I and class II events in the daytime are relatively small, while there are greater differences at night. Meanwhile, in class III, the RMSE performance at night is better than that in the daytime, but the COR is worse than that in the daytime. Therefore, it can be concluded that there are obvious and diversified differences among the simulation results shown by various PBL schemes under different types of near-surface wind events.

## 4  Summary and conclusions

In this study, a horizontal resolution of 0.33 km which is within the PBL gray zone resolution is employed to investigate the performance of four commonly used PBL schemes on near-surface wind simulation over the Sichuan Basin. In China, the near-surface wind prediction over Sichuan Basin has always a low score, and the main focus of wind simulation is about the pollutant diffusion under stable weather conditions at a horizontal resolution equals or greater than 1 km. Thus, we chose the site of Guanghan Airport as the representation, and conducted a total of 112 WRF sensitivity experiments, specifically focusing on 28 events with near-surface winds exceeding 6 m s$^{-1}$ by varying the PBL scheme, and assessed the impact of different PBL schemes on wind speed and direction simulations. Subsequent analyses considered factors such as diurnal variation of near-surface wind processes and circulation background to gain further understanding of their influence on model sensitivity. Therefore, the findings of our study offer the valuable insights in this region.

From our evaluation and analysis, the sensitivity of near-surface wind direction over Sichuan Basin to the four commonly used PBL schemes is very low, and the performance of MYNN2 is the worst when simulating the near-surface wind direction, while the other three schemes are generally consistent with the observations, and the MYJ scheme is the best for simulating NNE and NE winds. Our findings on wind direction is agree with the finding in many other researches (Gómez-Navarro et al., 2015; Tan et al., 2017; Shen and Du, 2023).

Generally speaking, no scheme can simulate the trend and wind speed of near-surface wind events well at the same time, which is also mentioned by Cohen et al. (2015). However, the 1.5-order QNSE local closure approximation scheme appears to be the best for the temporal variation, while MYJ is the scheme with smallest simulation error on wind speed. As the metrics RMSE and BIAS shows the similar characteristics, K-means cluster analysis is employed based on the COR and RMSE ,and the simulation results are divided into three categories. The first category of events showed poor correlation but small RMSE; the second category of events

showed high correlation but large RMSE; the third category of events showed high correlation coefficient and small RMSE. Further analysis found that the four PBL schemes can simulate the ground wind events caused by the typical strong cold air (occurring at 6:00-8:00 UTC), and there is little difference between them. For the near-surface wind events occurring in the midnight to early morning, they are mainly

concentrated in the second category; while the evening to night and the southerly wind process are mainly concentrated in the first category.

Therefore, multiple cases studies and K-means clustering analysis gives us the hint that the simulation performance of the PBL schemes mainly depends on the prevailing weather conditions of each case, such as circulation backgrounds and the

620 time of near-surface wind events. The results also point to the need for future research to explore the mechanisms behind the observed differences in wind speed simulation, particularly during nighttime and different atmospheric conditions.

. *Code and data availability.* The Weather Research and Forecasting (WRF)
model version 4.3.1 used in this study is freely available online and can be downloaded from https://www2. mmm.ucar.edu/wrf/users/download/get_source.html (Skamarock et al., 2008). The ERA5 data are available from ECMWF (https:// www.ecmwf.int/en/forecasts/datasets/reanalysis-datasets/era5, last access: 23 June
2023, DOI: https://doi.org/10.24381/cds.bd0915c6, Hersbach et al., 2018). The topographic data are available from Shuttle Radar Topography Mission (SRTM) 90 m DEM Digital Elevation Database (https://srtm.csi.cgiar.org/, last access: 20 June 2023).The observations and model output upon which this work is based are available from Zenodo (https://doi.org/10.5281/zenodo.11328605, Wang et al., 2024), and the
data can also be obtained from pwd@cafuc.end.cn.

*Author contributions.* QW conceptualized the study and conducted the simulations. BZ, YY and GC analyzed the model results, and QW and BZ contributed to the interpretations. The original draft of the paper was written by QW, and all the authors
took part in the edition and revision of it.

*Competing interests.* The contact author has declared that none of the authors has any competing interests.

*Disclaimer.* Publisher's note: Copernicus Publications remains neutral with regard to jurisdictional claims in published maps and institutional affiliations.

*Acknowledgments.* The authors acknowledge NCAR for the WRF model and ECMWF for the ERA5 reanalysis datasets.

*Financial support.* This research has been supported by the Joint Funds of the National Natural Science Foundation of China (grant no. U2242202), the National Key Research and Development Program of China (grant no. 2023YFC3007502 and 2022YFC3003902), the National Natural Science Foundation of China (grant no. 42030611), Major science and technology project of the Xizang Autonomous Region (grant no. XZ202402ZD0006), Sichuan Science and Technology Program (grant no. 2025ZNSFSC0334, 2022NSFSC0021 and 2023NSFSC0904).

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
