# Peer review of "Simulation performance of planetary boundary layer schemes in WRFV4.3.1 for near-surface wind over the western Sichuan Basin: a single site assessment"

_EGUsphere, 2024_

## Referee Comment (RC2)

**Review of "Simulation performance of different planetary boundary layer schemes in WRF V4.3.1 on wind field over Sichuan Basin within "Gray zone" resolution" by Want et. al.**

This study performed sensitivity experiments using four PBL schemes over the complex terrain in Sichuan Basin at the "Gray zone" resolution. The results show that while wind direction can be well reproduced and is not very sensitive to the PBL schemes, wind speed shows more sensitivity. The QNSE scheme had the best performance in reproducing the temporal variation, whereas the MYJ scheme had the smallest model bias. Using K-means classification, the authors concluded that the performance of the schemes is influenced by circulations. Predicting near-surface winds has practical importance and remains an ongoing challenge, especially over complex terrains. The choice of PBL has a significant impact on model performance. Therefore, this study is significant in this regard. However, the present form of analysis can be improved. I would overall recommend a major revision before it can be considered for possible publication.

**Major comments**

1. Since the authors emphasize this is a case study, one would expect case-by-case analysis. However, most analyses focus on bulk statistics or aggregate the data in some ways. The cases were selected solely based on wind speed exceeding 6 m/s. Is there any reason why this threshold is used? The length of each case should also be clarified.

2. The distribution probability analysis is a good way to evaluate the bulk features. How are the two parameters used in the Weibull distribution function connected to the distribution properties? Is the 10-min or event average used in the Weibull analysis? Please clarify.

3. The performance of PBL schemes can be influenced by many factors such as model assumptions, weather conditions, and local stability. Events with similar statistical errors do not directly reflect that they resulted from similar driving factors. Instead of classifying the events based on their statistical errors, I would suggest the opposite approach – classify the weather conditions and link the model errors to them.

**Specific comments**
Line 110-113: Please elaborate on why it is important to run the model at the "gray zone" resolutions?
Line 45: change to "winds"
Line 69: Please add WRF version.
Line 83 and other places: Add a space between the number and units.
Line 105-107: Please add reference to this statement.
Line 126-127: Why the case study is novel?
Line 133: Please replace "*" with "×".
Line 167: Change to "model configuration".
Table 1: What surface scheme was used?
Line 189: Why is 6 m/s selected as a threshold to select the cases? How long does a case last, a day?
Line 208: Suggest using Bias which is more commonly used.

Figure 2: What do the shading mean and dashed line mean? I assuming the dashed line is the threshold, which is 6 m/s in the text, but 5 m/s is showing in the figure. Please clarify. Again, from this figure, many of the cases were associated with diurnally varying winds while some cases were not. It would be interesting to see what synoptic scale/local conditions drive those wind patterns, and evaluate the PBL schemes' performance associated with those conditions.

Line 286: Please list some examples for the studies.

Line 290: Assuming the mean and median were calculated over the events. Please clarify.

Line 303-304: Please clarify that the "median" of the MJY ME is 0.96 m/s.

Line 307-319: This doesn't explain why MYJ is better in mean metrics while QNSE is better in variation. Since there is a suspicion that the performance of the PBL schemes differs under different stabilities, I'd suggest calculating the statistical metrics over different stabilities.

Line 330: Change to "10 m".

Figure 8: Looks like some points belonging to Cluster 1 is more close to the centroid of Cluster 2?

---

## Referee Comment (RC3)

"Simulation performance of different planetary boundary layer schemes in WRF V4.3.1 on wind field over Sichuan Basin within "Gray zone" resolution"

The authors undertake a "gray zone" WRF simulation campaign in an understudied region of China (Sichuan Basin) using different PBL schemes compared to one airport meteorological measurement. Results using common statistical error metrics for wind speed and direction are shown, where results for the different PBL schemes show good agreement for wind direction but poor agreement for wind speed. A k-means clustering technique is leveraged to help group different error metrics together and gauge PBL performance.

Overall, while I appreciate the study the authors are trying to undertake, I feel the analysis is underwhelming in breadth and justifications for modeling choices made are weak. Only one observation site is chosen for comparison, and yet it is believed to be representative of the entire region. Additionally, I am still left questioning why such a high spatial resolution WRF simulation was conducted, especially when the comparison was only performed against one observation site. Discussion of relevant meteorological phenomena is vague. For example, stability is often mentioned and used to understand the results, but no mention of a stability metric is used or referenced.

2 Data and Methods – general comments/questions

What is the temporal output of the WRF data, and how often is the model updated? I don't think this is every mentioned.

Why use such a high-resolution inner domain? Is it to prove that such simulations are possible with a mesoscale model in this region? It is unclear why such a high resolution WRF simulation is performed, especially when only considering one measurement site.

Only one reference measurement is used, yet strong claims are made about PBL scheme performance for just one 10 m wind tower measurement.

2 Data and Methods – specific comments/questions

pg. 5, line 166: A spin-up period of 3-hours is short, especially with a domain with complex topography. What was the reason for such a short spin-up time? I'm concerned this could affect results for the case studies, at least in the first few hours after spin-up are thrown out.

3 Overview of historical cases and evaluation of simulations results – general comments/questions

Throughout the results stability is mentioned many times by the authors, but it is never made clear how stability is defined in this study. If a discussion of model results compared to observations is going to take place, stability needs to be defined and/or referenced.

3 Overview of historical cases and evaluation of simulation results – specific comments/questions

pg. 8, line 240: A more thorough description of the dominate atmospheric circulations for each event is needed. Where is the "cold air" coming from? Is it a frontal passage, low-level jet, local terrain flows, etc.? Just saying "cold air" is not informative.

Figure 2b: I appreciate and understand what the authors are trying to convey here, as trying to plot 28 different time-series in one plot is not easy. I would emphasize in the figure caption though that this is not a continuous time-series, as upon first glance, the figure can be misleading. Also, what is the significance of the 5 m/s dotted line?

Figure 3: Why is the color bar range for wind speed values different than those of Figure 2a? This makes it difficult to compare observations and model results. It would be more beneficial visually if the observational wind rose from Figure 2 is combined into one figure with the model results of Figure 3 to more easily compare.

pg. 11, line 285: Again, it's hard to compare the differences in wind speed with a different color bar range and not having the plots side-by-side. Additionally, what are these other studies showing similar results? Cite them at the very least, and perhaps include some number ranges for reference.

pg. 14, line 356: Quantitatively state what these deviations are instead of using qualitative language. This advice goes for the entire paper, where qualitative statements are often more common than quantitative.

Figure 7: There is a lot of information being shown here, which is tricky to do, but would this be better as a line plot where each line is a different PBL scheme, and the error bars are shading around those lines? That might be easier to read than ~100 bar charts.

pg. 15, line 390: Perhaps the wrong word is being used here, but if the authors are going to make claims of significance, the authors should back up this statement with statistical significance tests. Otherwise, remove this statement and/or reword this sentence.

pg. 16, line 406: Unclassified results? What does this mean?

pg. 16, line 414: Are seasonal results not shown because there are no obvious seasonal differences?

---

## Author Comment (AC2)

**Response to the RC1**

(egusphere-2024-1532)

We sincerely appreciate the referee for these valuable comments on our manuscript entitled "Simulation performance of different planetary boundary layer schemes in WRF V4.3.1 on wind field over Sichuan Basin within "Gray zone" resolution"(egusphere-2024-1532). These comments are all valuable and helpful for improving our article. According to the referee's comments, we have made extensive modifications to our manuscript to make our results convincing. In the following point-to-point response, the comments from RC1 are in black font, and our replies are in blue font. We hope our answers clarify the referee's concerns.

**General comments:**

The manuscripts describes the results from WRF simulations over the Sichuan Basin, because the wind modelling is poor over this area. This is a interesting topic that warrants further investigations. My main concern with the paper is that the description of the measurements is missing. Effects of flow distortion on wind speed can be significant and should be described thoroughly. Technical specifications of the cup anemometer are not given. What is the observational uncertainty of the measurements? Are they regularly calibrated in a wind tunnel? Is a calm threshold applied for the wind vane? Furthermore, the local terrain effects will usually dominate wind speeds and direction measured at 10 m, which are not described at all in the manuscript (what is the local surface roughness etc?). In addition, there is unclear descriptions (see for example comment related to classification of "cold air" and "deep convection") and smaller technical issues. The authors have to convince the reader that the measurements are suitable for addressing a certain scientific question and relate the simulations to the specific research question. Finding the PBL scheme that can 'best' represent the wind distribution at one mast, is not so useful if a mast a kilometer away would lead to completely different results.

Response:

Thank you for your valuable comments and insightful suggestions on our manuscript. The wind direction and speed instruments installed at Guanghan Airport are primarily used to collect wind field data in support of flight operations. The International Civil Aviation Organization (ICAO) imposes stringent requirements on the collection, calibration, and quality control of meteorological data to ensure the accuracy and precision of wind measurements. Detailed information regarding wind measurement can be found in the following response to "Specific Issues". With regard to the question the referee mentioned at the end, we acknowledge the limitations highlighted by the referee, particularly the potential variations in results across different locations, which is a crucial consideration in wind speed simulation research. Nevertheless, we would like to elucidate the rationale and significance of our study from the following perspectives.

1. Simulations of wind speed at single site are frequently utilized to validate the performance

of numerical models in numerous scenarios(Denis et al.,2020). By selecting representative sites with high-quality observational data, valuable references can be provided for enhancing and optimizing PBL schemes. Besides, the observations from a number of stations are compared to the model output of wind speed and direction at the nearest grid point to each station (Gómez et al.,2015).

2. In practical applications, single-site wind speed simulations are frequently employed to fulfill specific engineering requirements. In such contexts, accurately simulating the wind speed distribution at a critical location holds direct practical significance.

3. Our objective in this study is not to ascertain a universally optimal PBL scheme applicable across all regions, but rather to assess the efficacy of different PBL schemes in specific locations within distinct geographic and climatic contexts, for instance, the western Sichuan Basin, and strong wind processes. This approach not only facilitates a more profound comprehension of the constraints and benefits associated with particular schemes, but also furnishes essential foundational data for subsequent multi-site or regional investigations.

References:

Gómez-Navarro, J. J., Raible, C. C., and Dierer, S.: Sensitivity of the WRF model to PBL parametrisations and nesting techniques: evaluation of wind storms over complex terrain, Geosci. Model Dev., 8, 3349–3363, https://doi.org/10.5194/gmd-8-3349-2015, 2015.

Denis, E.K., Muyiwa, S. A.:A preliminary sensitivity study of Planetary Boundary Layer parameterisation schemes in the weather research and forecasting model to surface winds in coastal Ghana,Renewable Energy, 146, 66-86,https://doi.org/10.1016/j.renene.2019.06.133, 2020.

**Specific issues:**

L72-L88: For each case study one can find a PBL scheme that does better than the rest. This section should also describe the physical process that cause a certain PBL scheme to do better and should be related the research question in this study.

Response:

Thank you very much for your valuable suggestion. Accordingly, we have revised this part in our manuscript as follows:

 "In China, Ma et al. (2014) conducted a series of sensitivity simulations on spring strong wind events in Xinjiang Province using the YSU, MYJ, and ACM2 schemes. The results indicated that the YSU scheme exhibited greater downward transport of high-level momentum, attributed to enhanced turbulent mixing effects. This improvement helps simulate temperature and moisture profiles more accurately during the daytime when convection is dominant (Hong et al., 2006). The YSU scheme has also been shown to be the optimal PBL scheme for

simulating 10-meter wind speeds in other regions (Cui et al., 2018; Li et al., 2018). However, in coastal areas like Fujian Province (Yang et al., 2014), studies have demonstrated that the MYJ scheme is the best choice for simulating near-surface wind speeds due to its advancements in calculating turbulent kinetic energy (TKE). The MYJ scheme computes TKE at each level, allowing for a more precise representation of turbulence within the boundary layer, which enhances its ability to model the generation, dissipation, and transport of turbulence (Janjié, 1990; Jaydeep et al., 2024). In the mountainous terrain of Huanghan and Guizhou, ACM2 has demonstrated superior performance in simulating near-surface wind speeds (Zhang and Yin, 2013; Mu et al., 2017). From these studies, it is evident that the performance of a PBL scheme is highly dependent on its ability to accurately represent the key physical processes within the boundary layer across different topographical contexts, leading to significant regional variations in the performance of PBL schemes in WRF."

L102: So the aim of the study is diffusion in stable cases: that should be moved earlier in the introduction and the discussion about the different PBL schemes should be related to it.

Response:

Thanks for your comment, and we apologize for the confusion caused by the sentences here.

In the matter of fact, we bring up the issue of pollutant dispersion here, aiming to emphasize that numerous studies hitherto have concentrated on the pollutant dispersion under stable and weak wind conditions in the Sichuan Basin, but less attention paid to unstable or strong wind events.

We have rewritten this sentence as below:

"Therefore, wind is not still as wildly studied as temperature and precipitation in Sichuan Basin, and numerous studies hitherto have concentrated on the pollutant dispersion under stable and weak wind conditions here, and less attention paid to unstable or strong wind events."

L114: add reference for "grey zone", e.g. https://journals.ametsoc.org/view/journals/atsc/61/14/1520-0469_2004_061_1816_tnmitt_2.0.co_2.xml

Response:

We kindly thanks for your suggestion, we have added this reference in the revised manuscript.

L218: I have never seen the formulas before so at least a reference should be provided. In general, the Weibull A and k should be found by fitting the Weibull distribution to the observed frequency histogram of the wind speeds.

Response:

Thanks for pointing out this.

Indeed, the probability density function (PDF) of the Weibull distribution can be expressed in various forms (Lai et al., 2006). In our manuscript, we calculated the PDF of the Weibull distribution following the approach of Jiang et al. (2015). In response to your comments, we have added two additional references to the revised manuscript.
References:

Lai, C. D., Murthy, D., and Xie, M. : Weibull Distributions and Their Applications, Springer Handbook of Engineering Statistics, Chapter 3. 63-78, 10.1007/978-1-84628-288-1_3, 2006.

Jiang, H., Wang, J. Z. , Dong, Y., Lu, H. :Comprehensive assessment of wind resources and the low-carbon economy: An empirical study in the Alxa and Xilin Gol Leagues of inner Mongolia, China, Renewable and Sustainable Energy Reviews, 50, 1304-1319, https://doi.org/10.1016/j.rser.2015.05.082, 2015.

Section 3: I am   missing description of the measurements: what kind of cup anemometer was being used? What kind of wind vane? Was any quality assurance done to make sure the data were adequate for this study. If you are measuring at 10 m the wind speed is totally dominated by the roughness length at the site, so that should be thoroughly described and assessed.

Response:

Thank you for your comment.

We fully understand your concern. In response, we have provided detailed information about the cup anemometer in Section 2 of our manuscript.

The wind direction and speed measurements were conducted using the FIRST CLASS three-cup anemometer and wind vane (Figure 1), manufactured by Thies Clima in Germany. The anemometer consists of three cups made from carbon fiber-reinforced plastic, which rotate in response to wind flow. This rotation is photoelectrically scanned and converted into a square wave signal, with the frequency of the signal being directly proportional to the rotation speed.

The wind vane's dynamic characteristics are optimized by its lightweight aluminum structure. The combined action of the wind vane and its counterweight results in a high damping coefficient and minimal delay distance, both of which contribute to the vane's excellent overall performance. The relevant technical specifications are provided in the following table.

Table 1 Technical specifications for cup anemometer and wind vane

| Technical Specifications | Description |
| --- | --- |

| | Cup Anemometer | Wind Vane |
|---|---|---|
| Ranege | 0.3-75m/s | 0-360° |
| Starting threshold | <0.3m/s | < 0.5 m/s at 10° amplitude (in accordance with ASTM D 5366-96) |
| | | < 0.2 m/s at 90° amplitude (in accordance with VDI 3786 part 2) |
| Accuracy | 1% of the measured value or < 0.2 m/s | 0.5° |
| Resolution | 0.05 m/s | 0.35° |

[Figure]

Figure 1 FIRST CLASS three-cup anemometer(left) and wind vane(right)

The anemometer and wind vane have undergone calibration twice a year. Quality control measures such as outlier removal are applied to the collected wind data. Statistical methods to detect outliers or unusual patterns in the data are applied in our study.

Regarding the roughness length mentioned by the referee, we acknowledge the significance of ground roughness length in wind speed research, particularly in weak wind conditions. However, our study primarily focuses on strong wind processes in the Sichuan Basin, where the impact of ground roughness is relatively minimal. Furthermore, we utilized consistent wind direction and speed observation data to assess various PBL schemes, ensuring uniformity and

mitigating variable discrepancies attributed to ground roughness. This approach centers on evaluating the performance of PBL schemes themselves(Yu et al., 2023). We believe this methodology can effectively gauge PBL scheme performance during strong wind processes while maintaining analytical simplicity and efficacy. However, it is worth noting that further exploration into ground roughness as a factor holds merit, especially when studying diverse wind speed conditions. Should future research necessitate an examination of PBL scheme performance across varying wind speeds, inclusion of ground roughness analysis will be considered.

According to the referee's comments, we have added the sentence in section 2.1 line 172 as follows:

"The terrain here is flat and homogeneous, and prevailing wind direction are north and northeast in climatology."

L239: A classification should classify a certain variable or process. But cold air is a property of the air, whereas deep convection is related to atmospheric stability. For example, you can have deep convection in very cold air. So this classification does not make sense.

Response:

Well, we really appreciate the referee's insightful comments.

We totally agree with you, that the deep convection occurs in very cold air in some region, and even if the thunderstorm gale processes still have the participation of cold air too. So, we are very sorry for our inaccurate expression. What we intend to clarify here is that the strong winds is caused mainly by convective weather system or non-convective weather system.

Therefore, the term 'cold air' as used in Table 3 denotes the case which is generated by incursion of cold air from northern regions like Siberia or Mongolia in Sichuan Basin, in such process, the cold air forces the warmer air ahead of it to rise rapidly, meanwhile, sharp temperature drop and changes in humidity can be observed. The term 'deep convection' specifically denotes the strong wind cases primarily caused by convective weather systems, often accompanied by thunderstorm. In such cases, the vertical motion or convection is the dominant.

Since the main focus of this paper is not on the meteorological cause of strong wind events in the Sichuan basin, we provide a simplified classification method here to help understand the differences of the performance between various planetary boundary layer (PBL) schemes in simulating strong near-surface winds caused by different meteorological processes. We have been aware of this problem here, accordingly, We clarified the two terms in the manuscript.

**Technical corrections**

L9: remove unique or specify what you mean with unique and why it is unique.

Response:

We kindly thank you for your suggestion, the points are well taken, we removed the word "unique".

L11: change to "In this study, the Weather Research ..."

Response:

Thanks for pointing out this typo, we have made a correction in the revised manuscript.

L13: I would change to near-surface wind fields because at the surface there is by definition no wind.

Response:

You are right, many thanks for pointing out this incorrect expression, we corrected it in our manuscript.

L15: You mean multiple case studies? Not sure what a multi-case study is.

Response:

Sorry for the error. What we're trying to say is that the study was based on multiple case studies (28 cases), rather than just one case. We have made a correction in our manuscript.

L19: You mean that the wind speed is sensitive to the choice of PBL scheme? In that case rewrite this sentence to make this more clear.

Response:

We really appreciate the referee's comment. We checked this sentence carefully, and rewrote it in our revised manuscript as follows:

"The results demonstrate that the wind direction can be well reproduced, yet it is not as sensitive to the PBL scheme as the near-surface wind speed."

L26: tiny -> small

Response:

Thanks for pointing out this incorrect expression, we corrected it in the manuscript.

L26-29: "However, ... were opposite". This line is unclear, do you mean that the case studies where chosen to be mostly around the morning? If yes, why? What is a evening to evening process? The opposite of what?

Response:

We apologize for the confusion caused by the sentence in the abstract. There are a few key moments that we didn't express clearly. First, we want to clarify that the time appeared in this sentence means the time when observed wind speed exceeded 6 m s$^{-1}$. Because the 28 near-surface wind events were chosen with a criteria of the maximum wind speed greater than 6 m s$^{-1}$ (L130).

We revised the abstract sentence in our manuscript as follows:

  "However, the simulation results for strong winds occurring during the mid-night to early morning hours exhibit poor root mean square errors but high correlation coefficients, whereas for strong wind processes happening in the early to late evening hours and for southwesterly wind processes demonstrate the opposite pattern."

L28: COR, if this is correlation coefficient just write it out. In general, abbrevations should be minimized in the abstract, because the reader does not know what they are at this point.

Response:

Thank you for pointing out this, we have made correction in the manuscript.

L35: "as the most" -> why is it the most fundamental? Temperature or humidity can also be fundamental, rewrite.

Response:

Thank you for your comment, we totally accept and have rewritten this sentence as "Wind, as the one of fundamental natural phenomenon in the atmosphere,......" in our manuscript.

L49: I would say topographic and underlying surface are referring to the same concept. And thermal effects are also dynamic?

Response:

Thank you for your comment.

The terms "topography" and "underlying surface" both refer to aspects of the Earth's surface in many fields, but have distinct meanings and applications in simulation of meteorology.

Topography refers to the physical appearance of the natural features of an area of land, especially the shape of its surface, such as mountains, valleys, rivers, or craters

on the surface. We know that mountains, valleys, and other topographical features can channel and redirect winds, creating localized variations in wind speed and direction.

The underlying surface refers to the physical material and properties present beneath a specific area, including soil type, bedrock, and other substrata.For instance, the type and density of vegetation can affect wind flow near the surface by increasing friction and altering turbulence.

As for the second question, what we intend to express is that, the near-surface wind fields can be affected by the dynamic process such as atmospheric pressure gradients, thermodynamic process including temperature gradients from the aspect of meteorology.

Thererfore, considering your suggestion, we have rewritten this part in our manuscript as follows:

"Near-surface wind fields are influenced by a combination of various factors, including atmospheric dynamic and thermodynamic processes (such as pressure gradient force, temperature gradients, and so on), topography (such as geographical features, elevation), and underlying surface (such as vegetation, land use) (Zhang et al., 2021)."

L116: vortices -> eddies

Response:

Suggestion taken, thank you for your comment.

L132: resolution -> horizontal resolution.

Response:

Suggestion taken, thank you for your comment.

L133: grids -> grid cells

Response:

Thanks for pointing out this typo, we have made correction in the manuscript.

Ll134: 45 what?

Response:

We are sorry for the mistake, it should be 45 levels, and we have made correction in the manuscript.

L138: upstream? The prevailing wind direction hasn't been introduced so the reader doesn't know what is upstream or downstream.

Response:

Many thanks for this comment, we have added the prevailing wind direction information in the manuscript (section 2.1).

L152: reference missing for SRTM3

Response:

Thank you for the comment. We have added the reference below in our manuscript.

Farr, T.G., Rosen, P. A., Caro, E. R., Crippen, R., Duren, R.M., Hensley, S., Kobrick, M., Paller, M., Rodríguez, E., Roth, L., Seal, D.A., Shaffer, S.J., Shimada, J., Umland, J.W., Werner, M., Oskin, M.E., Burbank, D.W., and Alsdorf, D.E.: The Shuttle Radar Topography Mission, Reviews of Geophysics, 45, https://doi.org/10.1029/2005RG000183, 2007.

L160: high -> height

Response:

Thanks for pointing out this typo, we have made correction in the manuscript.

L198: surface -> near surface, see earlier comment

Response:

Thanks you for your comment, we have made correction in the manuscript.

L206: I miss the definition of the overbar/overline.

Response:

Thanks you for your comment , we have made correction in the manuscript.

Fig 2: seies -> series

Response:

Thanks for pointing out this typo, we have made correction in the manuscript.

L265: Using overestimate and underestimate in relation with wind direction is confusing. Rewrite to use a directional metric.

Response:

Thank you for your comment, we have rewritten this part according to your suggestion:

"Besides, it is also shown that the occurrence frequencies of the wind fields simulated by four PBL schemes in the NNE and NE directions are all relatively higher than observation, but for wind in NNW direction, the simulated frequencies are significantly lower..."

Moreover, we added the directional statistical metrics (ME, RMSE and circular COR) for simulated 10-m wind (Table 4).

Table 4. Statistical metrics for simulated 10-m wind direction.

|  | Average Wind Direction (degrees) | ME(degrees) | RMSE(degrees) | Circular COR |
| --- | --- | --- | --- | --- |
| Observations | 22.2 | | | |
| YSU | 33.3 | 12.1 | 57.8 | 0.37 |
| MYJ | 32.1 | 12.5 | 58.9 | 0.36 |
| MYNN2 | 36.9 | 14.2 | 61.3 | 0.33 |
| QNSE | 31.0 | 9.8 | 62.1 | 0.30 |

---

## Author Comment (AC3)

**Point-to-point Responses to RC2**

**Review of "Simulation performance of different planetary boundary layer schemes in WRF V4.3.1 on wind field over Sichuan Basin within "Gray zone" resolution" by Want et. al.**

This study performed sensitivity experiments using four PBL schemes over the complex terrain in Sichuan Basin at the "Gray zone" resolution. The results show that while wind direction can be well reproduced and is not very sensitive to the PBL schemes, wind speed shows more sensitivity. The QNSE scheme had the best performance in reproducing the temporal variation, whereas the MYJ scheme had the smallest model bias. Using K-means classification, the authors concluded that the performance of the schemes is influenced by circulations. Predicting near-surface winds has practical importance and remains an ongoing challenge, especially over complex terrains. The choice of PBL has a significant impact on model performance. Therefore, this study is significant in this regard. However, the present form of analysis can be improved. I would overall recommend a major revision before it can be considered for possible publication.

Response:
Thank you for your thoughtful review and constructive comments on our manuscript. We have carefully considered your comments and suggestions for improvement. we appreciate your recognition of the significance of our study and your recommendation for a major revision. We are committed to making the necessary improvements and will ensure that the revised manuscript addresses all your comments comprehensively. Thank you once again for your valuable feedback, which will undoubtedly strengthen our work. Below are our point-by-point replies.

**Major comments**

1. Since the authors emphasize this is a case study, one would expect case-by-case analysis. However, most analyses focus on bulk statistics or aggregate the data in some ways. The cases were selected solely based on wind speed exceeding 6 m/s. Is there any reason why this threshold is used? The length of each case should also be clarified.

Response:

Thank you for your insightful comment.

We acknowledge the importance of conducting a case-by-case analysis, particularly in the context of a case study, as it allows for a deeper understanding. We will add analysis of different individual weather conditions in our manuscript.

However, it is true that the performance of a PBL scheme in simulating near-surface winds based on a single case study has inherent limitations.The success of a single case study might depend on specific initial conditions, boundary conditions, and external forcing factors. These conditions can affect the performance of the PBL scheme. So, we think it is necessary to conduct multiple simulations covering different time periods and meteorological conditions to evaluate the performance of a PBL scheme at a single site.

For the present study, we primarily focus on bulk statistics and aggregate data to provide an initial broad assessment across multiple cases. This approach offers a comprehensive overview of the model's performance under varying conditions, which we believe is essential for establishing a robust baseline.We appreciate your suggestion, and it aligns well with the next phase of our research, where we will focus on in-depth case-by-case analyses to further refine our understanding of the model's capabilities and limitations over western Sichuan Basin.

The threshold of 6 m/s for wind speed was selected based on its relevance to operational forecasting needs, especially in regions where wind speed significantly influences aviation safety and efficiency. This threshold aligns with established criteria in similar studies and ensures that the cases considered are both meteorologically significant and operationally relevant.

Regarding the length of each case, given that there are no instances of average wind speeds exceeding 6 m/s lasting longer than 24 hours in the western Sichuan Basin, we have therefore chosen a 24-hour length for each case.

2. The distribution probability analysis is a good way to evaluate the bulk features. How are the two parameters used in the Weibull distribution function connected to the distribution properties? Is the 10-min or event average used in the Weibull analysis? Please clarify.

Response:

Thank you for your insightful comments. We will incorporate these clarifications and ensure that the revised version addresses your concerns. Thank you for your valuable feedback.

 The two parameters of the Weibull distribution, typically referred to as the shape parameter ($k$) and the scale parameter ($\lambda$), play crucial roles in defining the distribution's characteristics. The shape parameter ($k$) indicates the distribution's variability and can provide insights into the nature of the wind speed distribution (e.g., whether it is more uniform or skewed). A value of $k < 1$ suggests that the distribution has a decreasing hazard function, while $k > 1$ indicates an increasing hazard function. The scale parameter ($\lambda$) provides a measure of the distribution's scale, representing the characteristic wind speed. We will expand the discussion in the revised manuscript to clarify how these parameters relate to the bulk features being analyzed.

In our analysis, the 10-min winds were used in the Weibull analysis. This choice was made to ensure that the results accurately reflect the temporal variability of the wind speeds and provide a robust representation of the data. We will clarify this point in the revised manuscript to ensure that readers understand the basis for our analysis.

3. The performance of PBL schemes can be influenced by many factors such as model assumptions, weather conditions, and local stability. Events with similar statistical errors do not directly reflect that they resulted from similar driving factors. Instead of classifying the events based on their statistical errors, I would suggest the opposite approach – classify the weather conditions and link the model errors to them.

Response:

Thank you for your valuable comments.

We appreciate your suggestion and recognize its potential relevance, we will consider incorporating a discussion of how future research could integrate both approaches—classifying events by weather conditions and analyzing statistical errors—to provide a more comprehensive understanding of PBL scheme performance over Sichuan Basin.

While we recognize that the performance of PBL schemes is indeed influenced by various factors, our primary objective in this study is to evaluate the performance of model outputs specifically in relation to statistical errors (RMSE and COR). Classifying events based on their statistical errors can help us distinguish between different cases that may have similar RMSE but different trends COR, revealing patterns in the performance of the model. For instance, a high RMSE with a low COR might indicate systematic errors rather than random fluctuations, suggesting specific adjustments may be needed in model parameters.This will provide a basis for subsequent mechanism analysis of these 28 individual simulations based on weather conditions and stability.

Therefore, we believe that our approach is suitable for the objectives of this study and provides valuable insights into the sensitivity of PBL schemes to statistical errors.

Thank you once again for your thoughtful suggestions, which will help enhance the clarity of our manuscript, and we are open to any further suggestions you may have.

**Specific comments**

Line 110-113: Please elaborate on why it is important to run the model at the "gray zone" resolutions?

Response:

Thank you for your valuable comments and suggestions.

The advancement of numerical models to a resolution of 1 km is a significant achievement, reflecting the current state-of-the-art in operational weather forecasting. However, as we look to the future, pushing towards even higher resolutions, becomes increasingly critical. Running models at these "gray zone" resolutions is essential for several reasons. For example, there is a need for more refined spatiotemporal resolution prediction of wind fields in many engineering applications. We will clarify in our manuscript.

Line 45: change to "winds".

Response:

Suggestion taken, thank you for your comment.

Line 69: Please add WRF version.

Response:

Suggestion taken, thank you for your comment.

Line 83 and other places: Add a space between the number and units.

Response:

Thank you for your comments and suggestions. We acknowledge this typo, and we appreciate your input.

However, based on the recommendations of another reviewer, we have incorporated a more detailed mechanistic analysis. As a result, we have revised the original statement, which you will see in the updated manuscript.

Line 105-107: Please add reference to this statement.

Response:

Suggestion taken, thank you for your comment.

Line 126-127: Why the case study is novel?

Response:

Thank you for your comments.

In the original manuscript, Line 126-127: "In this study, the experimental approach is different from what has been used in other studies, where one case or long continuous time is simulated."

So far, there is a  substantial amount of research employing the WRF model to simulate wind fields, predominantly concentrating on one case study or use continuous long-run, large-scale regional assessments, or evaluations at single station. However, studies that assess WRF simulations through multiple case studies (Gómez et al., 2015) remain relatively scarce in the literature.

As we stated in the above response, to evaluate the performance of a PBL scheme over a large area, it is usually necessary to conduct multiple simulations covering different time periods and meteorological conditions. This approach provides a broader data sample, ensuring that the evaluation results have statistical significance, rather than being based on the outcomes of a single case.

Line 133: Please replace "*" with "×".

Response:

Suggestion taken, thank you for your comment.

Line 167: Change to "model configuration" .

Response:

Thanks for pointing out this typo, we have made correction in the manuscript.

Table 1: What surface scheme was used?

Response:

Thank you for pointing this out. We will update Table 1 to include the surface layer scheme and the surface physic scheme used in our simulations to provide greater clarity and completeness of the methodology.

Table 1 PBL schemes and surface layer scheme

| PBL scheme | Surface layer scheme |
| --- | --- |
| YSU | Revised MM5 Monin-Obukhov scheme |
| MYJ | MYJ |
| MYNN2 | MYNN |
| QNSE | QNSE |

Line 189: Why is 6 m/s selected as a threshold to select the cases? How long does a case last a day?

Response:

Thanks for your comment regarding the selection of the 6 m/s threshold. The choice of this particular threshold is rooted in its significance to aviation operations, which is the primary focus of this study. Specifically, wind speeds around 6 m/s are critical for aircraft during takeoff and landing, where maintaining control and ensuring safety are of utmost importance.

By selecting 6 m/s as the threshold, we aimed to ensure that the cases analyzed in this study are directly relevant to these crucial phases of flight. This choice enables the study to provide findings that are not only scientifically rigorous but also practically applicable to the field of aviation weather, thereby enhancing the relevance of the results to real-world aviation scenarios.

We hope this explanation clarifies the rationale behind our decision, and we are open to any further suggestions you may have.

Line 208: Suggest using Bias which is more commonly used.

Response:

Suggestion taken, thank you for your comment.

Figure 2: What do the shading mean and dashed line mean?I assuming the dashed line is the threshold, which is 6 m/s in the text, but 5 m/s is showing in the figure. Please clarify. Again, from this figure, many of the cases were associated with diurnally varying winds while some cases were not. It would be interesting to see what synoptic scale/local conditions drive those wind patterns, and evaluate the PBL schemes' performance associated with those conditions.

Response:

Thank you for your insightful comments regarding Figure 2, and we'll add some clarification here.

We appreciate your observations and would like to clarify a few points.

The shading in the figure was employed to highlight the time series of the 28 selected cases, which are discontinuous across days. To enhance clarity for the readers, we shaded every alternate case in pink. Regarding the dashed line, to facilitate understanding and to avoid confusion, we opted to include a contour line at 5 m/s. This choice was made because we selected individual cases where the 10-minute average wind speed was greater than or equal to 6 m/s. Since displaying all 28 cases with frequency of every 10 minutes in detail was unfeasible, we presented hourly data, which may inadvertently suggest that some individual cases did not reach the 6 m/s threshold. Therefore, we draw a dashed line of 5m/s instead of 6m/s in the picture to increase the readability of the picture .

It is important to note that in the Chengdu Plain areas of the western Sichuan Basin, the average wind speed and climate conditions are typically below 3 m/s. Nonetheless, it is possible for the 10-minute average wind to exceed 6 m/s induced by synoptic regimes.

We also appreciate your suggestion to investigate the synoptic scale and local conditions that drive the observed diurnal variability in wind patterns. We plan to conduct a detailed analysis of the errors associated with these 28 cases in relation to weather conditions in a future publication. This analysis will provide valuable insights for improving wind field predictions and refining PBL schemes in the region.

Once again, thank you for your valuable feedback. We will strive to incorporate your suggestions as we move forward with our research.

Line 286: Please list some examples for the studies.

Response:

Thank you for your valuable comment. We have added the references in our manuscript at line 286.

Line 290: Assuming the mean and median were calculated over the events. Please clarify.

Response:

Thank you for your valuable comment.

In our manuscript, Figure 4 contains box plots for 28 cases, each showing the RMSE (Root Mean Square Error), COR (Correlation), and Bias of simulations from four different PBL schemes compared to observations at 10-minute intervals. So, the mean and median correlation coefficients mentioned in the sentence were calculated across 28 events for each of the four PBL schemes.

A box plot is a powerful statistical tool used to visualize the distribution and central tendencies of data. The line in the middle of the box represents the median, which gives us insight into the central tendency of the data. For instance, the median of RMSE can indicate the typical size of the model's prediction error. By observing the shape of the box and the length of the whiskers, one can determine the symmetry of the data and whether there is skewness.

As shown in the figure, both the mean and median values for all schemes fall within the range of 0.4 to 0.6. This indicates a tendency for the coefficients to cluster around this range across the events.

In figure 4, the box plots and heat maps are both given, box plots offer an in-depth analysis of individual cases, while heatmaps provide a global overview through color gradients. By combining these two, it is possible to simultaneously analyze both local details and global trends, helping to identify specific issues in individual cases and overall trends.According to you insightful comment,, we have revised this part to make it more clear.

We hope our response addresses your concerns. Please let us know if further clarification is needed.

Line 303-304: Please clarify that the "median" of the MYJ ME is 0.96 m/s.

Response:

Thank you for your comment.

The median value of 0.96 m/s for the MYJ scheme was obtained from Figure 4c (as shown in the picture given below), which presents a box plot of the mean errors (ME) calculated from 28 cases simulated by the four PBL schemes. The box plot provides a visual summary of the distribution of these errors, including the median, quartiles.

In Figure 4c, the median line within the MYJ box corresponds to a value of 0.96 m/s, which represents the central tendency of the mean errors across the 28 cases for this specific PBL scheme. This figure was generated using the error data from each case, where the box plot effectively illustrates the spread and central values of the mean errors.

We hope this explanation clarifies how the median value was derived, and we will revised this part in our manuscript. Please let us know if further details are needed.

[Figure]

Figure1 Captured from Figure 4 c in original manuscript.

Line 307-319: This doesn't explain why MYJ is better in mean metrics while QNSE is better in variation. Since there is a suspicion that the performance of the PBL schemes differs under different stabilities, I'd suggest calculating the statistical metrics over different stabilities.

Response:

Thank you for your insightful comments and suggestions.

We agree that examining the performance of the PBL schemes under different stability conditions could provide valuable insights into their respective strengths and weaknesses. To address your suggestion, we plan to calculate the statistical metrics for four PBL schemes across different stability conditions. This analysis will help clarify the reasons behind the observed performance differences and enhance the robustness of our findings.We will incorporate this additional analysis into our revised manuscript.

Line 330: Change to "10 m".

Response:

Suggestion taken, thank you for your comment.

Figure 8: Looks like some points belonging to Cluster 1 is more close to the centroid of Cluster 2?

Response:

Thank you for your observation regarding the K-means clustering results.

While it may visually seem that some points belong to another cluster when viewed in a simplified space, the assignment was determined by their position in the complete feature space, where the relationship between features can be more complex.

Additionally, it's important to note that the centroids themselves are calculated based on the mean position of all points within each cluster, and slight overlaps or close proximities between clusters can occur, especially if the clusters are not well-separated.

---

## Author Response (AR1)

Response to referee comments

*We sincerely appreciate the referees for their valuable and insightful comments on our manuscript entitled "Simulation performance of different planetary boundary layer schemes in WRF V4.3.1 on wind field over Sichuan Basin within 'Gray zone' resolution" (egusphere-2024-1532). The feedback provided by the referees has proven to be instrumental in enhancing the quality and clarity of our research.*

*These comments are not only valuable but also serve as a critical resource for improving various aspects of our article, including methodology, data interpretation, and overall presentation. We have taken each comment seriously and conducted a thorough review of our manuscript to ensure that we address all concerns raised by the referees comprehensively.*

*In response to the three referee's comments, we have made extensive modifications throughout our manuscript aimed at strengthening the validity and reliability of our results. This includes refining our analytical approach, clarifying ambiguous sections, providing additional context where necessary, and ensuring that all figures and tables accurately represent the findings discussed.*

*This response document provides a detailed account of the changes implemented in relation to each specific comment from the referees. For ease of reference, referee comments are presented in black font while author responses are highlighted in blue font. It is important to note that all line numbers mentioned correspond directly to locations within the revised version of the manuscript.*

Referee 1: Referee Comments
**General comments:**

The manuscripts describes the results from WRF simulations over the Sichuan Basin, because the wind modelling is poor over this area. This is a interesting topic that warrants further investigations. My main concern with the paper is that the description of the measurements is missing. Effects of flow distortion on wind speed can be significant and should be described thoroughly. Technical specifications of the cup anemometer are not given. What is the observational uncertainty of the measurements? Are they regularly calibrated in a wind tunnel? Is a calm threshold applied for the wind vane? Furthermore, the local terrain effects will usually dominate wind speeds and direction measured at 10 m, which are not described at all in the manuscript (what is the local surface roughness etc?). In addition, there is unclear descriptions (see for example comment related to classification of "cold air" and "deep convection") and smaller technical issues. The authors have to convince the reader that the measurements are suitable for addressing a certain scientific question and relate the simulations to the specific research question. Finding the PBL scheme that can 'best' represent the wind distribution at one mast, is not so useful if a mast a kilometer away would lead to completely different results.

Response:

Thank you for your valuable comments and insightful suggestions on our manuscript. The wind direction and speed instruments installed at Guanghan Airport are primarily used to collect wind field data in support of flight operations. The International Civil Aviation Organization (ICAO) imposes stringent requirements on the collection, calibration, and quality control of meteorological data to ensure the accuracy and precision of wind measurements. Detailed information regarding wind measurement can be found in the following response to "Specific Issues". With regard to the question the referee mentioned at the end, we acknowledge the limitations highlighted by the referee, particularly the potential variations in results across different locations, which is a crucial consideration in wind speed simulation research. Nevertheless, we would like to elucidate the rationale and significance of our study from the following perspectives.

1. Simulations of wind speed at single site are frequently utilized to validate the performance of numerical models in numerous scenarios(Denis et al.,2020). By selecting representative sites with high-quality observational data, valuable references can be provided for enhancing and optimizing PBL schemes. Besides, the observations from a number of stations are compared to the model output of wind speed and direction at the nearest grid point to each station (Gómez et al.,2015).

2. In practical applications, single-site wind speed simulations are frequently employed to fulfill specific engineering requirements. In such contexts, accurately simulating the wind speed distribution at a critical location holds direct practical significance.

3. Our objective in this study is not to ascertain a universally optimal PBL scheme applicable across all regions, but rather to assess the efficacy of different PBL schemes in specific locations within distinct geographic and climatic contexts, for instance, the western Sichuan Basin, and strong wind processes. This approach not only facilitates a more profound comprehension of the constraints and benefits associated with particular schemes, but also furnishes essential foundational data for subsequent multi-site or regional investigations.

References:

Gómez-Navarro, J. J., Raible, C. C., and Dierer, S.: Sensitivity of the WRF model to PBL parametrisations and nesting techniques: evaluation of wind storms over complex terrain, Geosci. Model Dev., 8, 3349–3363, https://doi.org/10.5194/gmd-8-3349-2015, 2015.

Denis, E. K., Muyiwa, S. A.: A preliminary sensitivity study of Planetary Boundary Layer parameterisation schemes in the weather research and forecasting model to surface winds in coastal Ghana,Renewable Energy, 146, 66-86, https://doi.org/10.1016/j.renene.2019.06.133, 2020.

**Specific issues:**

L72-L88: For each case study one can find a PBL scheme that does better than the rest. This section should also describe the physical process that cause a certain PBL scheme to do better and should be related the research question in this study.

Response:

Thank you very much for your valuable suggestion. Accordingly, we have revised this part in our manuscript as follows:

Line 71-95:    In China, Ma et al. (2014) conducted a series of sensitivity simulations on spring strong wind events in Xinjiang Province using the YSU, MYJ, and ACM2 schemes. The results indicated that the YSU scheme exhibited greater downward transport of high-level momentum, attributed to enhanced turbulent mixing effects. This improvement helps simulate temperature and moisture profiles more accurately during the daytime when convection is dominant (Hong et al., 2006). The YSU scheme has also been shown to be the optimal PBL scheme for simulating 10-meter wind speeds in other regions (Cui et al., 2018; Li et al., 2018). However, in coastal areas like Fujian Province (Yang et al., 2014), studies have demonstrated that the MYJ scheme is the best choice for simulating near-surface wind speeds due to its advancements in calculating turbulent kinetic energy (TKE). The MYJ scheme computes TKE at each level, allowing for a more precise representation of turbulence within the boundary layer, which enhances its ability to model the generation, dissipation, and transport of turbulence (Janjié, 1990; Jaydeep et al., 2024). In the mountainous terrain of Huanghan and Guizhou, ACM2 has demonstrated superior performance in simulating near-surface wind speeds (Zhang and Yin, 2013; Mu et al., 2017). From these studies, it is evident that the performance of a PBL scheme is highly dependent on its ability to accurately represent the key physical processes within the boundary layer across different topographical contexts, leading to significant regional variations in the performance of PBL schemes in WRF.

L102: So the aim of the study is diffusion in stable cases: that should be moved earlier in the introduction and the discussion about the different PBL schemes should be related to it.

Response:

Thanks for your comment, and we apologize for the confusion caused by the sentences here.

In the matter of fact, we bring up the issue of pollutant dispersion here, aiming to emphasize that numerous studies hitherto have concentrated on the pollutant dispersion under stable and weak wind conditions in the Sichuan Basin, but less attention paid to unstable or strong wind events.

We have rewritten this sentence as below:

Line 104-107: Therefore, wind is not still as wildly studied as temperature and precipitation in Sichuan Basin, and numerous studies hitherto have concentrated on the pollutant dispersion under stable and weak wind conditions here, and less attention paid to unstable or strong wind

events.

L114: add reference for "grey zone", e.g. https://journals.ametsoc.org/view/journals/atsc/61/14/1520-0469_2004_061_1816_tnmitt_2.0.co_2.xml

Response:

We kindly thanks for your suggestion, we have added this reference in the revised manuscript.

Line 120: in numerical forecasting (Wyngaard, 2004; Liu et al., 2018; Yu et al., 2022).

L218: I have never seen the formulas before so at least a reference should be provided. In general, the Weibull A and k should be found by fitting the Weibull distribution to the observed frequency histogram of the wind speeds.

Response:

Thanks for pointing out this.

Indeed, the probability density function (PDF) of the Weibull distribution can be expressed in various forms (Lai et al., 2006). In our manuscript, we calculated the PDF of the Weibull distribution following the approach of Jiang et al. (2015). We have added two additional references to the revised manuscript.

Line 236: distribution of wind speed (Lai, et al., 2006; Jiang, et al., 2015).

References:

Lai, C. D., Murthy, D., and Xie, M. : Weibull Distributions and Their Applications, Springer Handbook of Engineering Statistics, Chapter 3. 63-78, 10.1007/978-1-84628-288-1_3, 2006.

Jiang, H., Wang, J. Z. , Dong, Y., Lu, H. :Comprehensive assessment of wind resources and the low-carbon economy: An empirical study in the Alxa and Xilin Gol Leagues of inner Mongolia, China, Renewable and Sustainable Energy Reviews, 50, 1304-1319, https://doi.org/10.1016/j.rser.2015.05.082, 2015.

Section 3: I am missing description of the measurements: what kind of cup anemometer was being used? What kind of wind vane? Was any quality assurance done to make sure the data were adequate for this study. If you are measuring at 10 m the wind speed is totally dominated by the roughness length at the site, so that should be thoroughly described and assessed.

Response:

Thank you for your comment.

**For the measurements:**

We fully understand your concern. In response, we have provided detailed description about the cup anemometer in Section 2 of the revised manuscript, from line 168-180.

In our research, the wind direction and speed measurements were conducted using the FIRST CLASS three-cup anemometer and wind vane (Figure 1), manufactured by Thies Clima inc. in Germany. The anemometer consists of three cups made from carbon fiber-reinforced plastic, which rotate in response to wind flow. This rotation is photoelectrically scanned and converted into a square wave signal, with the frequency of the signal being directly proportional to the rotation speed.

The wind vane's dynamic characteristics are optimized by its lightweight aluminum structure. The combined action of the wind vane and its counterweight results in a high damping coefficient and minimal delay distance, both of which contribute to the vane's excellent overall performance. The relevant technical specifications are provided in Table 1. The anemometer and wind vane have undergone calibration twice a year. Statistical methods to detect outliers or unusual patterns in the data are applied in our study.

Table 1 Technical specifications for cup anemometer and wind vane

| Technical Specifications | Description | |
| --- | --- | --- |
| | Cup Anemometer | Wind Vane |
| Ranege | 0.3-75m/s | 0-360° |
| Starting threshold | <0.3m/s | < 0.5 m/s at 10° amplitude (in accordance with ASTM D 5366-96) |
| | | < 0.2 m/s at 90° amplitude (in accordance with VDI 3786 part 2) |
| Accuracy | 1% of the measured value or < 0.2 m/s | 0.5° |
| Resolution | 0.05 m/s | 0.35° |

[Figure]

Figure 1 FIRST CLASS three-cup anemometer(left) and wind vane (right)

Based on the description above, the information is added as follows:

Line168-180: The terrain here is flat and homogeneous, and prevailing wind direction are north and northeast in climatology. Wind direction and speed were measured using the FIRST CLASS three-cup anemometer and wind vane, both manufactured by Thies Clima inc. in Germany. The anemometer has a measurement range of 0.3 to 75 m s$^{-1}$ and a starting threshold of less than 0.3 m s$^{-1}$, with an accuracy of 1% of the measured value or less than 0.2 m s$^{-1}$. The wind vane covers a measurement range of 0 to 360°, with a starting threshold of less than 0.5 m s$^{-1}$ at a 10° amplitude (as per ASTM D 5366-96) and 0.2 m s$^{-1}$ at a 90° amplitude (according to VDI 3786 Part 2), and an accuracy of 0.5°. During the research period, the anemometers were annually calibrated by accredited institutions. Before incorporating the wind data into our analysis, we performed basic data checks and quality control procedures, including outlier removal.

**Regarding the roughness length:**

We appreciate the referee's comments regarding surface roughness length, which is indeed an important factor in near-surface wind research. Numerous studies have demonstrated the significance of conducting sensitivity experiments with varying surface roughness lengths within the WRF model. However, our investigation primarily centers on strong wind events induced by synoptic systems at Guanghan Airport in the western Sichuan Basin. Given the flat terrain and the absence of significant obstacles surrounding the airport, the influence of surface roughness is relatively minimal. Consequently, we have opted to utilize the default roughness values provided by the WRF model in our analysis.

However, the primary objective of our study is to conduct sensitivity experiments related to the planetary boundary layer (PBL).To maintain consistency across all experimental setups, we

have opted to use the default surface roughness values provided by the WRF model. This approach allows us to more effectively evaluate the performance of different PBL schemes in simulating near-surface wind fields. By keeping the roughness values constant, we can focus on assessing the PBL schemes themselves.

We appreciate your understanding and hope this clarification addresses your concerns.

Nonetheless, we recognize that further investigation into the role of surface roughness is warranted, particularly in studies encompassing a broader range of wind speed conditions. Should future research necessitate an examination of PBL scheme performance across varying wind speeds, we will consider the incorporation of surface roughness analysis as a significant factor.

According to the referee's comments, we have added the sentence in section 2.1 as follows:

Line 168-169:The terrain here is flat and homogeneous, and prevailing wind direction are north and northeast in climatology.

L239: A classification should classify a certain variable or process. But cold air is a property of the air, whereas deep convection is related to atmospheric stability. For example, you can have deep convection in very cold air. So this classification does not make sense.

Response:

Well, we really appreciate the referee's insightful comments.

We totally agree with you, that the deep convection occurs in very cold air in some region, and even if the thunderstorm gale processes still have the participation of cold air too. So, we are very sorry for our inaccurate expression. What we intend to clarify here is that the strong winds is caused mainly by convective weather system or non-convective weather system.

Since the main focus of this paper is not on the meteorological cause of strong wind events in the Sichuan basin, we provide a simplified classification method here to help understand the differences of the performance between various planetary boundary layer (PBL) schemes in simulating strong near-surface winds caused by different meteorological processes. We have been aware of this problem here, accordingly, and have clarified the two terms in the revised manuscript.

Line 260-266: As for the dominated factors of each event, the term 'cold air' in Table 3 was used to denote the cases which are generated by incursion of cold air from northern regions like Siberia or Mongolia in Sichuan Basin, often accompanied by sharp temperature drop and changes in humidity. The term 'convective system' specifically denotes the strong wind cases primarily caused by convective weather systems, often accompanied by thunderstorm. In such cases, the vertical motion or convection is the dominant.

**Technical corrections**

L9: remove unique or specify what you mean with unique and why it is unique.

Response:

We kindly thank you for your suggestion, the points are well taken, we have removed the word "unique" at line 10.

L11: change to "In this study, the Weather Research ..."

Response:

Thanks for pointing out this typo, we have made a correction in the revised manuscript at line 12.

L13: I would change to near-surface wind fields because at the surface there is by definition no wind.

Response:

Thanks for your suggestion, we have corrected it in our manuscript.

L15: You mean multiple case studies? Not sure what a multi-case study is.

Response:

Sorry for the error. What we're trying to say is that the study was based on multiple case studies (28 cases), rather than just one case. We have made a correction in our revised manuscript at line 15.

L19: You mean that the wind speed is sensitive to the choice of PBL scheme? In that case rewrite this sentence to make this more clear.

Response:

We really appreciate the referee's comment. We checked this sentence carefully, and rewrote it in our revised manuscript as follows:

Line 18-20: The results demonstrate that the wind direction can be well reproduced, yet it is not as sensitive to the PBL scheme as the near-surface wind speed.

L26: tiny -> small

Response:

Thanks for pointing out this incorrect expression, we have corrected it in the revised manuscript at line 27.

L26-29: "However, ... were opposite". This line is unclear, do you mean that the case studies where chosen to be mostly around the morning? If yes, why? What is a evening to evening process? The opposite of what?

Response:

We apologize for the confusion caused by the sentence in the abstract. There are a few key moments that we didn't express clearly. First, we want to clarify that the time appeared in this sentence means the time when observed wind speed exceeded 6 m s$^{-1}$. Because the 28 near-surface wind events were chosen with a criteria of the maximum wind speed greater than 6 m s$^{-1}$ (line 130).

We have revised the abstract sentence in our manuscript as follows:

Line 27-31: However, the simulation results for strong winds occurring during the mid-night to early morning hours exhibit poor root mean square errors but high correlation coefficients, whereas for strong wind processes happening in the early to late evening hours and for southwesterly wind processes demonstrate the opposite pattern.

L28: COR, if this is correlation coefficient just write it out. In general, abbrevations should be minimized in the abstract, because the reader does not know what they are at this point.

Response:

Thank you for pointing out this, we have made a correction in the manuscript at line 29.

L35: "as the most" -> why is it the most fundamental? Temperature or humidity can also be fundamental, rewrite.

Response:

Thank you for your comment, we totally accept and have rewritten this sentence as "Wind, as the one of fundamental natural phenomenon in the atmosphere,......" at line 36 of the revised manuscript.

L49: I would say topographic and underlying surface are referring to the same concept. And thermal effects are also dynamic?

Response:

Thank you for your comment.

The terms "topography" and "underlying surface" both refer to aspects of the Earth's surface in

many fields, but have distinct meanings and applications in simulation of meteorology.

Topography refers to the physical appearance of the natural features of an area of land, especially the shape of its surface, such as mountains, valleys, rivers, or craters on the surface. We know that mountains, valleys, and other topographical features can channel and redirect winds, creating localized variations in wind speed and direction.

The underlying surface refers to the physical material and properties present beneath a specific area, including soil type, bedrock, and other substrata.For instance, the type and density of vegetation can affect wind flow near the surface by increasing friction and altering turbulence.

As for the second question, what we intend to express is that, the near-surface wind fields can be affected by the dynamic process such as atmospheric pressure gradients, and thermodynamic process such as temperature gradients from the aspect of meteorology.

Therefore, considering your suggestion, we have rewritten this part in our manuscript as follows:

Line 50-53: Near-surface wind fields are influenced by a combination of various factors (Zhang et al., 2021), including atmospheric dynamic and thermodynamic processes (such as pressure gradient force, temperature gradients, and so on), topography (such as geographical features, elevation), and underlying surface (such as vegetation, land use).

L116: vortices -> eddies

Response:

Suggestion taken, thank you for your comment.

L132: resolution -> horizontal resolution.

Response:

Suggestion taken, thank you for your comment.

L133: grids -> grid cells

Response:

Thanks for pointing out this typo, we have made a correction at line 141 in the revised manuscript.

Ll134: 45 what?

Response:

We are sorry for the mistake, it should be 45 levels, and we have corrected at line 141 in the revised manuscript.

L138: upstream? The prevailing wind direction hasn't been introduced so the reader doesn't know what is upstream or downstream.

Response:

Many thanks for this comment, we have added the prevailing wind direction information at line 169 in the revised manuscript (section 2.1).

L152: reference missing for SRTM3

Response:

Thank you for the comment. We have added the reference at line 159 in the revised manuscript.

Farr, T.G., Rosen, P. A., Caro, E. R., Crippen, R., Duren, R.M., Hensley, S., Kobrick, M., Paller, M., Rodríguez, E., Roth, L., Seal, D.A., Shaffer, S.J., Shimada, J., Umland, J.W., Werner, M., Oskin, M.E., Burbank, D.W., and Alsdorf, D.E.: The Shuttle Radar Topography Mission, Reviews of Geophysics, 45, https://doi.org/10.1029/2005RG000183, 2007.

L160: high -> height

Response:

Thanks for pointing out this typo, we have made correction at line 168 in the revised manuscript.

L198: surface -> near surface, see earlier comment

Response:

Thanks you for your comment, we have made a correction in the manuscript.

L206: I miss the definition of the overbar/overline.

Response:

Thanks you for your comment, we have corrected at line 226-229 in the revised manuscript.

Fig 2: seies -> series

Response:

Thanks for pointing out this typo, we have made a correction in the manuscript.

L265: Using overestimate and underestimate in relation with wind direction is confusing.

Rewrite to use a directional metric.

Response:

Thank you for your comment, we have rewritten this part according to your suggestion:

Line 330-333: Besides, it is also shown that the occurrence frequencies of the wind fields simulated by four PBL schemes in the NNE and NE directions are all relatively higher than observation, but for wind in NNW direction, the simulated frequencies are significantly lower...

Moreover, we added the directional statistical metrics (BIAS, RMSE and circular COR) for simulated 10-m wind (Table 4).

Table 4. Statistical metrics for simulated 10-m wind direction.

| | Average Wind Direction (°) | BIAS(°) | RMSE(°) | Circular COR |
|---|---|---|---|---|
| Observations | 22.2 | | | |
| YSU | 33.3 | 12.1 | 57.8 | 0.37 |
| MYJ | 32.1 | 12.5 | 58.9 | 0.36 |
| MYNN2 | 36.9 | 14.2 | 61.3 | 0.33 |
| QNSE | 31.0 | 9.8 | 62.1 | 0.30 |

Referee 2: Referee Comments

**Review of "Simulation performance of different planetary boundary layer schemes in WRF V4.3.1 on wind field over Sichuan Basin within "Gray zone" resolution" by Want et. al.**

This study performed sensitivity experiments using four PBL schemes over the complex terrain in Sichuan Basin at the "Gray zone" resolution. The results show that while wind direction can be well reproduced and is not very sensitive to the PBL schemes, wind speed shows more sensitivity. The QNSE scheme had the best performance in reproducing the temporal variation, whereas the MYJ scheme had the smallest model bias. Using K-means classification, the authors concluded that the performance of the schemes is influenced by circulations. Predicting near-surface winds has practical importance and remains an ongoing challenge, especially over complex terrains. The choice of PBL has a significant impact on model performance. Therefore, this study is significant in this regard. However, the present form of analysis can be improved. I would overall recommend a major revision before it can be considered for possible publication.

Response:

Thank you for your thoughtful review and constructive comments on our manuscript. We have carefully considered your comments and suggestions for improvement. We appreciate your recognition of the significance of our study and your recommendation for a major revision. We are committed to making the necessary improvements and will ensure that the revised manuscript addresses all your comments comprehensively. Thank you once again for your valuable feedback, which will undoubtedly strengthen our work. Below are our point-by-point replies.

**Major comments**

1. Since the authors emphasize this is a case study, one would expect case-by-case analysis.

However, most analyses focus on bulk statistics or aggregate the data in some ways. The cases were selected solely based on wind speed exceeding 6 m/s. Is there any reason why this threshold is used? The length of each case should also be clarified.

Response:

Thank you for your insightful comment.

We acknowledge the importance of conducting a case-by-case analysis, particularly in the context of a case study, as it allows for a deeper understanding.

However, it is true that the performance of a PBL scheme in simulating near-surface winds

based on a single case study has inherent limitations. The success of a single case study might depend on specific initial conditions, boundary conditions, and external forcing factors. These conditions can affect the performance of the PBL scheme. So, we think it is necessary to conduct multiple simulations covering different time periods and meteorological conditions to evaluate the performance of a PBL scheme at a single site.

For the present study, we primarily focus on bulk statistics and aggregate data to provide an initial broad assessment across multiple cases. This approach offers a comprehensive overview of the model's performance under varying conditions, which we believe is essential for more in-depth analysis in the future.We appreciate your suggestion, and it aligns well with the next phase of our research, where we will focus on in-depth case-by-case analyses to further refine our understanding of the model's capabilities and limitations over western Sichuan Basin.

The threshold of 6 m/s for wind speed was selected based on its relevance to operational forecasting needs, especially in regions where wind speed significantly influences aviation safety and efficiency. This threshold aligns with established criteria in similar studies and ensures that the cases considered are both meteorologically significant and operationally relevant.

Regarding the length of each case, given that there are no instances of average wind speeds exceeding 6 m/s lasting longer than 24 hours in the western Sichuan Basin, we have therefore chosen a 24-hour length for each case.

2. The distribution probability analysis is a good way to evaluate the bulk features. How are the two parameters used in the Weibull distribution function connected to the distribution properties? Is the 10-min or event average used in the Weibull analysis? Please clarify.

Response:

Thank you for your insightful comments regarding the distribution probability analysis and the Weibull distribution function. We will incorporate these clarifications and ensure that the revised version addresses your concerns. Thank you for your valuable feedback.

In our analysis, the 10-min winds were used in the Weibull analysis. We will clarify this point in the revised manuscript to ensure that readers understand the basis for our analysis.

We also have rewritten this part to connect two parameters used in the Weibull distribution function and the distribution properties as follows:

Line 382-405: Figure 5 shows the frequency distribution of different winds with the observed and the simulated wind data at Guanghan Airport. As can be seen, the observed wind speed distribution is left-skewed, primarily due to the concentration of wind speeds within the 1-4 m s$^{-1}$ range, where the cumulative frequency exceeds 0.6. When comparing the spread of each PBL scheme's distribution to the observations, all four PBL schemes exhibit a wider distribution, indicating overestimation of the wind speed variability.

In order to give a more precision comparison during four PBL schemes, the corresponding Weibull distribution fitting curve fitting curves, shape parameters, and scale parameters were calculated, as shown in Figure 5. The shape parameter ($k$) reflects the distribution of wind speeds. A lower $k$ value indicates a more dispersed distribution with greater wind speed variability, while a higher $k$ value suggests a more concentrated distribution with less variability. The observed shape parameter is 1.79, while the shape parameters for YSU, MYJ, MYNN2, and QNSE are 1.89, 1.83, 1.93, and 1.77, respectively. QNSE has a shape parameter very close to the observed value, indicating it simulates wind variability most similarly to the actual observations. From the shape parameter perspective, QNSE provides the most similar wind speed distribution to the observations. YSU and MYNN2 show more concentrated wind speed distributions, potentially underestimating wind speed variability. The observed scale parameter is 3.30 m s$^{-1}$, while the scale parameters for YSU, MYJ, MYNN2, and QNSE are 5.20 m s-1, 4.69 m s$^{-1}$, 4.88 m s$^{-1}$, and 5.25 m s$^{-1}$, respectively. In terms of the scale parameter, all PBL schemes overestimate wind speeds, with YSU and QNSE showing the largest deviations. MYJ and MYNN2 are closer to the observed wind speeds.

3. The performance of PBL schemes can be influenced by many factors such as model assumptions, weather conditions, and local stability. Events with similar statistical errors do not directly reflect that they resulted from similar driving factors. Instead of classifying the events based on their statistical errors, I would suggest the opposite approach – classify the weather conditions and link the model errors to them.

Response:

Thank you for your valuable comments. We appreciate your suggestion regarding the potential relevance of classifying events by weather conditions and analyzing statistical errors.

While we acknowledge that the performance of PBL schemes is influenced by various factors, the primary objective of our study is to evaluate model outputs specifically in relation to statistical errors, such as RMSE (Root Mean Square Error) and COR (Correlation coefficient). Classifying events based on their statistical errors allows us to differentiate between cases that may exhibit similar RMSE values but possess different trends in COR. This distinction can reveal important patterns in model performance. For example, a high RMSE coupled with a low COR might indicate systematic errors rather than random fluctuations, suggesting that specific adjustments to model parameters may be required. This approach will also provide a foundation for subsequent mechanistic analyses of the 28 individual simulations based on weather conditions and stability.

Therefore, we believe that our approach is well-suited to the objectives of this study and offers valuable insights into the sensitivity of PBL schemes to statistical errors. We will consider incorporating both approaches in future research to provide a more comprehensive understanding of PBL schemes performance over the western Sichuan Basin.

**Specific comments**

Line 110-113: Please elaborate on why it is important to run the model at the "gray zone" resolutions?

Response:

Thank you for your valuable comments and suggestions.

The advancement of numerical models to a resolution of 1 km is a significant achievement, reflecting the current state-of-the-art in operational weather forecasting. However, as we look to the future, pushing towards even higher resolutions, becomes increasingly critical. So, running models at these "gray zone" resolutions is essential for several reasons. For example, there is a need for more refined spatiotemporal resolution prediction of wind fields in many engineering applications.

We have revised this part in our manuscript as follows:

Line 110-120: However, there has been no comprehensive evaluation of the performance of PBL schemes in simulating the near-surface wind field over the Sichuan Basin, whether using a single measurement site or multiple regional sites. Thus, combing the spatiotemporal refinement requirements from low-altitude flight safety, this study aims to evaluate the performance of four commonly used PBL schemes in reproducing near-surface wind fields with high spatiotemporal resolution by using the wind data from Guanghan Airport in the western Sichuan Basin. So, a horizontal resolution of 0.3 km was used in the model set-up for research, which is a major challenge in such region, because the spatial resolution is in the range of 0.1-1km, which is often referred as "gray zone" in numerical forecasting (Wyngaard, 2004; Liu et al., 2018; Yu et al., 2022).

Line 45: change to "winds".

Response:

Suggestion taken, thank you for your comment.

Line 69: Please add WRF version.

Response:

Suggestion taken, thank you for your comment.

Line 83 and other places: Add a space between the number and units.

Response:

Thank you for your comments and suggestions. We have added in our revised manuscript.

Line 105-107: Please add reference to this statement.

Response:

Suggestion taken, thank you for your comment.

Turnipseed, A., Anderson, D., Burns, S., Blanken, P., and Monson, R.: Airflows and turbulent flux measurements in mountainous terrain: Part 2: Mesoscale effects, Agricultural and Forest Meteorology, 125, 187-205, 10.1016/j.agrformet.2004.04.007, 2004,

Rajput, A., Singh, N., Singh, J., and Rastogi S.: Insights of Boundary Layer Turbulence Over the Complex Terrain of Central Himalaya from GVAX Field Campaign. Asia-Pac J Atmos Sci 60, 143–164, https://doi.org/10.1007/s13143-023-00341-5, 2024.

Line 126-127: Why the case study is novel?

Response:

Thank you for your comments.

So far, there is a substantial amount of research employing the WRF model to simulate wind fields, predominantly concentrating on one case study or use continuous long-run for large-scale regional assessments, or evaluations at single station. However, studies that assess WRF simulations through multiple case studies remain relatively scarce in the literature.

As we stated in the above response, to evaluate the performance of a PBL scheme over a large area, it is usually necessary to conduct multiple simulations covering different time periods and meteorological conditions. This approach provides a broader data sample, ensuring that the evaluation results have statistical significance, rather than being based on the outcomes of a single case.

Line 133: Please replace "*" with " ×".

Response:

Suggestion taken, thank you for your comment.

Line 167: Change to "model configuration" .

Response:

Thanks for pointing out this typo, we have made correction in the revised manuscript.

Table 1: What surface scheme was used?

Response:

Thank you for pointing this out. We have updated Table 2 as follows:

Table 2 The four selected PBL schemes and surface schemes in experiment.

| PBL scheme | Advantages | Surface layer scheme | Land surface scheme |
|---|---|---|---|
| YSU | 1st-order closure scheme that is widely utilized for its robust representation of turbulence closure | Revised MM5 Monin-Obukhov scheme | Noah MP |

| | | | |
|---|---|---|---|
| | processes (Hong et. al., 2006). | | |
| MYJ | A 1.5-order closure scheme that is known for its effectiveness in capturing vertical mixing processes (Janjié, 1990). | MYJ | Noah MP |
| MYNN2 | A 1.5-order closure scheme that improves the simulation of sub-grid scale turbulence (Nakanishi and Niino , 2009). | MYNN | Noah MP |
| QNSE | A 1.5-order turbulence closure scheme that accounts for both turbulent and non-turbulent mixing processes in the atmosphere (Sukoriansky and Galperin, 2006). | QNSE | Noah MP |

Line 189: Why is 6 m/s selected as a threshold to select the cases? How long does a case last a day?

Response:

Thanks for your comment regarding the selection of the 6 m/s threshold. The choice of this particular threshold is rooted in its significance to low-level aviation operations, which is the primary focus of this study. Specifically, wind speeds around 6 m/s are critical for aircraft during takeoff and landing, where maintaining control and ensuring safety are of utmost importance.

By selecting 6 m/s as the threshold, we aimed to ensure that the cases analyzed in this study are directly relevant to these crucial phases of flight. This choice enables the study to provide findings that are not only scientifically rigorous but also practically applicable to the field of aviation weather, thereby enhancing the relevance of the results to real-world aviation scenarios.

The selection of individual cases is based on wind speed data at 10-minute intervals, when the wind speed greater than or equal to 6m/s last for 30 minutes. We have revised at line 212 in the revised manuscript.

Line 208: Suggest using Bias which is more commonly used.

Response:

Suggestion taken, thank you for your comment.

We have revised the formulas in the manuscript and substituted all occurrences of ME with BIAS. Additionally, Figure 4 has been adjusted accordingly.

In the revised manuscript, the revisions are located at line 230, 231, 234 and 374.

Figure 2: What do the shading mean and dashed line mean? I assuming the dashed line is the threshold, which is 6 m/s in the text, but 5 m/s is showing in the figure. Please clarify. Again, from this figure, many of the cases were associated with diurnally varying winds while some cases were not. It would be interesting to see what synoptic scale/local conditions drive those wind patterns, and evaluate the PBL schemes' performance associated with those conditions.

Response:

Thank you for your insightful comments regarding Figure 2, and we have added some clarification here.

We appreciate your observations and would like to clarify a few points.

The shading in the figure was employed to highlight the time series of the 28 selected cases, which are discontinuous across days. To enhance clarity for the readers, we shaded every alternate case in pink. Regarding the dashed line, we acknowledge your assumption that it represents a threshold of 6 m/s, as mentioned in the text. However, to facilitate understanding and to avoid confusion, we opted to include a contour line at 5 m/s. This choice was made because we selected individual cases where the 10-minute average wind speed was greater than or equal to 6 m/s. Since displaying all 28 cases with frequency of every 10 minutes in detail was unfeasible, we presented hourly data, which may inadvertently suggest that some individual cases did not reach the 6 m/s threshold. In consideration of the misleading results, we have replaced 5m/s contour line with 6m/s at Figure 2 and explained it in the paper.

It is important to note that in the Chengdu Plain areas of the western Sichuan Basin, the average wind speed and climate conditions are typically below 3 m/s. Nonetheless, it is possible for the average wind to exceed 6 m/s over a 10-minute interval influenced by synoptic regimes.

We also appreciate your suggestion to investigate the synoptic scale and local conditions that drive the observed diurnal variability in wind patterns. We plan to conduct a detailed analysis of the errors associated with these 28 cases in relation to weather conditions and atmospheric stability in a future publication. This analysis will provide valuable insights for improving wind field predictions and refining PBL schemes in the region.

Once again, thank you for your valuable feedback. We have revised this part in our manuscript as follows:

Line 273-283: The near-surface wind speed in the Sichuan Basin exhibits a distinct diurnal variation, characterized by lower wind speeds in the morning and evening and higher wind

speeds at midday. In order to analyze the temporal variation of wind speed under different conditions, the hourly time series of the observed wind speed for 28 cases is presented in Fig. 2. It is showed that many cases with the incursion of cold air exhibit diurnal variation characteristics. Because, in these cases, cold air predominantly affects the western Sichuan Basin around midday (Table 3). However, for strong wind events such as cases No. 9, 13, 25, and 26, which were caused by convective systems, there was no clear diurnal variation in wind speed, and is characterized by sudden changes in wind speed, reflecting the transient and localized nature of convective processes.

Line 286: Please list some examples for the studies.

Response:

Thank you for your valuable comment. We have revised this part:

Line 329- 336: In fact, by comparing Fig. 2 and Fig. 3, it seems that all the four PBL schemes exhibit obvious exaggeration of wind speed, which is also shown in other numerous studies (Dzebre et al.,2020; Ma et al., 2024). For instance, in the research by Yu et al. (2022), all 11 WRF PBL schemes overestimate near-surface wind speeds by approximately 1 m s$^{-1}$ in the Hebei Plain. Similarly, in the experiment conducted by Gómez et al. (2015), the MYJ scheme strongly overestimates the maximum wind speed by more than 10 m s$^{-1}$ at 50% of the locations, while the YSU scheme shows deviations greater than 3 m s$^{-1}$.

References:

Dzebre, D. E. and Muyiwa, S. A.: A preliminary sensitivity study of Planetary Boundary Layer parameterisation schemes in the weather research and forecasting model to surface winds in coastal Ghana, Renewable Energy, 146, 66-86, 2020.

Ma, Y.-F., Wang, Y., Xian, T., Tian, G., Lu, C., Mao, X., and Wang, L.-P.: Impact of PBL schemes on multiscale WRF modeling over complex terrain, Part I: Mesoscale simulations. Atmospheric Research, 297, https://doi.org/10.1016/j.atmosres.2023.107117, 2024.

Line 290: Assuming the mean and median were calculated over the events. Please clarify.

Response:

Thank you for your valuable comment.

In our manuscript, Figure 4 contains box plots for 28 cases, each showing the RMSE (Root Mean Square Error), COR (Correlation), and BIAS of simulations from four different PBL schemes compared to observations at 10-minute intervals. So, the mean and median correlation coefficients mentioned in the sentence were calculated across 28 events for each of the four PBL schemes.

As shown in the figure, both the mean and median values for all schemes fall within the range

of 0.4 to 0.6. This indicates a tendency for the coefficients to cluster around this range across the events.

In Figure 4, the box plots and heat maps are given, box plots offer an in-depth analysis of individual cases, while heatmaps provide a global overview through color gradients. By combining these two, it is possible to simultaneously analyze both local details and global trends, helping to identify specific issues in individual cases and overall trends. According to you insightful comment, we have revised this part to make it more clear.

Line 337-341: To further evaluate the advantages and disadvantages of each scheme in simulating near-surface wind speed, three statistical metrics (COR, RMSE and BIAS) were calculated. These statistics were derived from data recorded at 10-minute intervals across 28 distinct events, as illustrated in Figure 4.

We hope our response addresses your concerns. Please let us know if further clarification is needed.

Line 303-304: Please clarify that the "median" of the MYJ ME is 0.96 m/s.
Response:

Thank you for your comment.

The median value of 0.96 m/s for the MYJ scheme was obtained from Figure 4c (as shown in the picture given below), which presents a box plot of the mean errors (ME) calculated from 28 cases simulated by the four PBL schemes. The box plot provides a visual summary of the distribution of these errors, including the median, quartiles.

In Figure 4c, the median line within the MYJ box corresponds to a value of 0.96 m/s, which represents the central tendency of the mean errors across the 28 cases for this specific PBL scheme. This figure was generated using the error data from each case, where the box plot effectively illustrates the spread and central values of the mean errors.

We hope this explanation clarifies how the median value was derived, and we have revised this part in our manuscript. Please let us know if further details are needed.

Line 354-358: The BIAS is consistent with RMSE as illustrated in the Fig. 4 (c), except that the median and mean BIAS is not as close as RMSE shows in MYJ scheme, indicating that the systematic error (BIAS) might be either too high or too low in certain cases. However, overall, MYJ scheme is highly precise and has little variance in its performance, which is crucial for accurate weather forecasts.

[Figure]

Figure1 Captured from Figure 4 c in original manuscript.

Line 307-319: This doesn't explain why MYJ is better in mean metrics while QNSE is better in variation. Since there is a suspicion that the performance of the PBL schemes differs under different stabilities, I'd suggest calculating the statistical metrics over different stabilities.

Response:

Thank you for your insightful comments and suggestions.

We agree that calculating statistical metrics over different atmospheric stabilities would provide deeper insights into the performance of the PBL schemes. However, due to the lack of wind tower data at close proximity, we are unable to calculate key stability parameters such as the Richardson number and Monin-Obukhov length, which would allow for a precise classification of stability regimes.

In our research, we believe that the classification of simulations into daytime and nighttime periods in our study already provides a meaningful differentiation of atmospheric stability conditions. Daytime periods typically correspond to more unstable atmospheric stratification, while nighttime periods are usually characterized by stable conditions. By analyzing the model performance across these diurnal cycles, we have, to some extent, addressed the PBL schemes' ability to simulate near-surface wind under different stability conditions.

We hope this approach is acceptable and are open to further suggestions on improving the analysis given the available data constraints.

Line 330: Change to "10 m".

Response:

Suggestion taken, thank you for your comment.

Figure 8: Looks like some points belonging to Cluster 1 is more close to the centroid of

Cluster 2?

Response:

Thank you for your observation regarding the K-means clustering results.

In our analysis, the classification was performed using the full set of features, and the algorithm assigned points to clusters based on their overall proximity to the centroids in this higher-dimensional space. While it may visually seem that some points belong to another cluster when viewed in a simplified space, the assignment was determined by their position in the complete feature space, where the relationship between features can be more complex.

Additionally, it's important to note that the centroids themselves are calculated based on the mean position of all points within each cluster, and slight overlaps or close proximities between clusters can occur, especially if the clusters are not well-separated.

Referee 3: Referee Comments

Title: Simulation performance of different planetary boundary layer schemes in WRFV4.3.1 on wind field over Sichuan Basin within "Gray zone" resolution
No.:egusphere-2024-1532

The authors undertake a "gray zone" WRF simulation campaign in an understudied region of China (Sichuan Basin) using different PBL schemes compared to one airport meteorological measurement. Results using common statistical error metrics for wind speed and direction are shown, where results for the different PBL schemes show good agreement for wind direction but poor agreement for wind speed. A k-means clustering technique is leveraged to help group different error metrics together and gauge PBL performance.

Overall, while I appreciate the study the authors are trying to undertake, I feel the analysis is underwhelming in breadth and justifications for modeling choices made are weak. Only one observation site is chosen for comparison, and yet it is believed to be representative of the entire region. Additionally, I am still left questioning why such a high spatial resolution WRF simulation was conducted, especially when the comparison was only performed against one observation site. Discussion of relevant meteorological phenomena is vague. For example, stability is often mentioned and used to understand the results, but no mention of a stability metric is used or referenced.

Response:

Thank you for your detailed and constructive feedback. We really appreciate your acknowledgment of the study's intentions and understand the concerns you've raised regarding the scope of the analysis, the justification for modeling choices, and the depth of the discussion.

As the referee mentioned in the general comments that only one observation site is chosen for comparison, and yet it is believed to be representative of the entire region. We are sorry that the scope of our title is too large and the text does not explain clearly. We also realize that this will cause confusion to readers, so we have revised our title as "Simulation performance of planetary boundary layer schemes in WRFV4.3.1 for near-surface wind over the western Sichuan Basin: a single site assessment" and further explain the purpose and area of this study in the introduction.

We appreciate your feedback and are committed to improving the manuscript accordingly. We believe our revisions will address the concerns raised and enhance the overall quality and rigor of our study. Below are our point-by-point replies, which we hope will address your comments satisfactorily.

2 Data and Methods – general comments/questions

What is the temporal output of the WRF data, and how often is the model updated? I don't think this is every mentioned.

Why use such a high-resolution inner domain? Is it to prove that such simulations are possible with a mesoscale model in this region? It is unclear why such a high resolution WRF simulation is performed, especially when only considering one measurement site.

Only one reference measurement is used, yet strong claims are made about PBL scheme performance for just one 10 m wind tower measurement.

Response:

Thank you for your insightful feedback.

Firstly, we are very sorry for our negligence of the temporal output of the WRF data in our manuscript. The temporal output of our WRF model is 10 minutes. We have added this information in the revised manuscript. However, in the original text, the information "the model is updated every 3 hours" was given at line 163.

Secondly, we appreciate your concern regarding the use of a high-resolution inner domain in our simulations. The decision to employ high-resolution simulations, specifically for the inner domain of the model, is motivated by the critical requirements of airport meteorological support services rather than solely demonstrating the feasibility of such simulations. The advancement of numerical models to a resolution of 1 km is a significant achievement, reflecting the current state-of-the-art in operational weather forecasting. However, as we look to the future, pushing towards even higher resolutions, becomes increasingly critical. Running models at these "gray zone" resolutions is essential for several reasons. For example, there is a need for more refined spatiotemporal resolution prediction of wind fields in many engineering applications.We have clarified in our revised manuscript.

Currently in the inner domain, it is difficult to collect stations with the open access wind data at a time resolution of 10 minutes, so our research is focused on a single station, which limits our ability to generalize findings across different terrains and climates. However, when we use one single site, we also chose the approach that conduct multiple simulations covering different time periods and meteorological conditions to evaluate the performance of a PBL scheme. We believe that this approach not only strengthens the reliability of our findings but also demonstrates the feasibility and advantages of high-resolution mesoscale modeling in such complex and unique environments. Nonetheless, we acknowledge that further studies incorporating multiple sites would be beneficial to validate and expand upon these results.

What's more, single site research is also seen in many other studies, such as Mantovani Júnior et al.(2023). In their research, the performance of eight PBL schemes is evaluated using detailed observations of the 2014 and 2015 dry season periods, specifically from 30 September to 2 October 2014, as well as from 14 to 16 October 2015.The observational data were collected at a research site named T3 (3.213°S, 60.598° W, 50 m) located nearby the confluence of the Negro and Solimoes rivers in Manacapuru City, Amazonas, Brazil, the result show that the local MYNN2.5 scheme showed the overall best performance for PBLH prediction, mainly at night.

In the research of Draxl et al.(2012), one coastal site over western Denmark was used to evaluate the wind speed and vertical wind shears simulated by different PBL schemes of WRF model. Dong et al.(2018) used the observation data from an one Arctic coastal meteorology station (named Tiksi Station) to evaluate high-resolution WRF simulations of strong surface wind for the Arctic region.

Moving forward, we plan to increase the number of observational sites to provide a more comprehensive analysis of the gray zone resolution issues across a broader region, including complex basins. This will enable us to refine our models further and address the resolution challenges specific to basin topographies.

References:

Mantovani Júnior, J.A.; Aravéquia, J.A.; Carneiro, R.G.; Fisch, G. Evaluation of PBL Parameterization

Schemes in WRF Model Predictions during the Dry Season of the Central Amazon Basin. Atmosphere, 14, 850, https://doi.org/10.3390/atmos14050850,2023.

Draxl, C., Hahmann, A., Peña, A.,and Giebel, G.: Evaluating winds and vertical shear from Weather Research and Forecasting model forecasts using seven planetary boundary layer schemes. Wind Energy. 17. 10.1002/we.1555, 2014.

Dong, H. T., Cao, S. Y., Takemi, T., and Ge, Y. Y.: WRF simulation of surface wind in high latitudes, Journal of Wind Engineering and Industrial Aerodynamics, 179, 287-296, https://doi.org/10.1016/j.jweia.2018.06.009,2018.

2 Data and Methods – specific comments/questions

pg. 5, line 166: A spin-up period of 3-hours is short, especially with a domain with complex topography. What was the reason for such a short spin-up time? I'm concerned this could affect results for the case studies, at least in the first few hours after spin-up are thrown out.

Response:

Thank you for your valuable feedback regarding the spin-up period of 3 hours.

In the matter of fact, There is a lack of consensus and clear guidance on identifying the suitable length of spin-up time, the optimal spin-up time vary by event and situation. We acknowledge that the WRF spin-up time should not be too short as it is hard to develop the appropriate atmospheric circulations, but not the longer the better (Liu et al., 2023). The proper spin-up time depends on the time needed for initialization, which can be affected by the size of the domain and the local boundary perturbations (Warner et al., 1997; Kleczek et al., 2014), most studies have empirically chosen 6-12 hours as the spin-up time. However, their research area is large and the spatial resolutions is 1 km or lower, lacking the finer granularity of the present study. Empirically, the finer the grid size, the more time steps it would have in a given time window, hence faster spin-up. Given that our case studies primarily focus on short-term phenomena, and the model has a high spatial resolution of 0.3 km, and the analysis is based on 10-minute intervals, we choose a spin up time of 3hours.

To ensure that the short spin-up period did not adversely affect the outcomes of our study, we conducted sensitivity tests by extending the spin-up duration and comparing results (Figure 1 and Table 1). The differences in key output variables were minimal, which supports the adequacy of the 3-hour spin-up time for our specific application.

[Figure]

Figure 1. The wind rose for observation and simulated near-surface wind field corresponding to the four PBL schemes when considering spin up time of 6 hours, the circles represent the relative frequency (%), and the colors represent wind speed.

Table 1 Comparison of metrics for four PBL schemes between spin-up time of 3 hours and 6hours.

| PBL Scheme | Metric | Spin-up time | |
| --- | --- | --- | --- |
| | | 3h | 6h |
| YSU | RMSE | 2.62 | 2.84 |
| | COR | 0.60 | 0.58 |
| | Bias | 1.63 | 1.85 |
| MYJ | RMSE | 2.28 | 2.46 |
| | COR | 0.58 | 0.56 |
| | Bias | 1.18 | 1.36 |
| MYNN2 | RMSE | 2.41 | 2.62 |
| | COR | 0.54 | 0.51 |
| | Bias | 1.34 | 1.51 |

| QNSE | RMSE | 2.75 | 3.01 |
|------|------|------|------|
|      | COR  | 0.61 | 0.59 |
|      | Bias | 1.70 | 1.97 |

Reference:

Liu, Y., Zhuo, L., Han, D. W.: Developing spin-up time framework for WRF extreme precipitation simulations, Journal of Hydrology, 620, https:// doi.org/ 10.1016/ j.jhydrol. 2023.129443, 2023.

Warner, T. T., Peterson, R. A., and Treadon, R. E.: A tutorial on lateral boundary conditions as a basic and potentially serious limitation to regional numerical weather prediction, B. Am. Meteorol. Soc., 78, 2599–2617, 1997.

Kleczek, M. A., Steeneveld, G. J., and Holtslag, A. A.: Evaluation of the weather research and forecasting mesoscale model for GABLS3: impact of boundary-layer schemes, boundary conditions and spin-up, Bound.-Lay. Meteorol., 152, 213–243, 2014.

3 Overview of historical cases and evaluation of simulations results – general comments/questions.

Throughout the results stability is mentioned many times by the authors, but it is never made clear how stability is defined in this study. If a discussion of model results compared to observations is going to take place, stability needs to be defined and/or referenced.

Response:

Thank you for highlighting the need for clarity regarding the concept of "stability" mentioned throughout the results section.

The stable mentioned in line 307-312, which means the stability of boundary layer. However, the stability mentioned in line 380-382, what we want to express through data analysis is that the error distribution obtained from the simulation of 28 individual cases by QNSE scheme has small changes, unlike the large changes in the error simulation of other schemes.We have replaced stable with a more accurate description in our revised manuscript.

Line 450-455: The QNSE scheme shows little variation in its simulation results during the daytime and the best simulation ability at noon across 28 different wind cases. The consistent performance suggests the reliable outputs for various strong wind events occurring within the daytime. In contrast, during nighttime simulations, there is a increase in variability among the results produced by the QNSE scheme. Overall, the performance of the PBL schemes varies based on the time of day, indicating that the PBL schemes may be sensitive to diurnal changes in atmospheric conditions.

3 Overview of historical cases and evaluation of simulation results – specific comments/questions

pg. 8, line 240: A more thorough description of the dominate atmospheric circulations for each event is needed. Where is the "cold air" coming from? Is it a frontal passage, low- level jet, local terrain flows, etc.? Just saying "cold air" is not informative.

Response:

We agree with the referee's suggestions and have incorporated the recommended changes into the manuscript.

Line 260-265: As for the dominated factors of each event, the term 'cold air' in Table 3 was used to denote the cases which are generated by incursion of cold air from northern regions like Siberia or Mongolia in Sichuan Basin, often accompanied by sharp temperature drop and changes in humidity. The term 'convective system' specifically denotes the strong wind cases primarily caused by convective weather systems, often accompanied by thunderstorm.

Figure 2b: I appreciate and understand what the authors are trying to convey here, as trying to plot 28 different time-series in one plot is not easy. I would emphasize in the figure caption though that this is not a continuous time-series, as upon first glance, the figure can be misleading. Also, what is the significance of the 5 m/s dotted line?

Response:

Thank you for your insightful feedback. We have revised the figure caption of Figure 2.

The shading in the figure was employed to highlight the time series of the 28 selected cases, which are discontinuous across days. To enhance clarity for the readers, we shaded every alternate case in pink. Regarding the dashed line, to facilitate understanding and to avoid confusion, we opted to include a contour line at 5 m/s. This choice was made because we selected individual cases where the 10-minute average wind speed was greater than or equal to 6 m/s. Since displaying all 28 cases with frequency of every 10 minutes in detail was unfeasible, we presented hourly data, which may inadvertently suggest that some individual cases did not reach the 6 m/s threshold. Therefore, we draw a dashed line of 5m/s instead of 6m/s in the picture. In consideration of the misleading results, we have replaced 5m/s contour line with 6m/s at Figure 4 and explained it in the paper.

Figure 3: Why is the color bar range for wind speed values different than those of Figure 2a? This makes it difficult to compare observations and model results. It would be more beneficial visually if the observational wind rose from Figure 2 is combined into one figure with the model results of Figure 3 to more easily compare.

Response:

We agree with the referee's suggestions and have incorporated the recommended changes into the manuscript.We believe these changes enhance the overall clarity and comparability of the data and hope they address your concerns.

[Figure]

(a) Observation          (b) YSU Sheme          (c) MYJ Sheme

(d) MYNN2 Sheme          (e) QNSE Sheme

**Figure 3.** The wind rose for observation and simulated near-surface wind field corresponding to the four PBL schemes, the circles represent the relative frequency (%), and the colors represent wind speed.

pg. 11, line 285: Again, it's hard to compare the differences in wind speed with a different color bar range and not having the plots side-by-side. Additionally, what are these other studies showing similar results? Cite them at the very least, and perhaps include some number ranges for reference.

Response:

Thank you for your feedback. To address your concern, we have revised the figures to ensure a consistent color bar range across all wind speed plots. Furthermore, we have arranged the plots side-by-side within a single figure. We also have revised this part as follows:

Line 329-336: In fact, by comparing Fig. 2 and Fig. 3, it seems that all the four PBL schemes exhibit obvious exaggeration of wind speed, which is also shown in other numerous studies (Dzebre et al.,2020; Ma et al., 2024;). For instance, in the research by Yu et al. (2022), all 11 WRF PBL schemes overestimate near-surface wind speeds by approximately 1 m s$^{-1}$ in the Hebei Plain. Similarly, in the experiment conducted by Gómez et al. (2015), the MYJ scheme strongly overestimates the maximum wind speed by more than 10 m s-1 at 50% of the locations, while the YSU scheme shows deviations greater than

3 m s$^{-1}$.

We hope these revisions adequately address your concerns and improve the clarity and relevance of our work.

References:

Yu, E., Bai, R., Chen, X., and Shao, L.: Impact of physical parameterizations on wind simulation with WRF V3.9.1.1 under stable conditions at planetary boundary layer gray-zone resolution: a case study over the coastal regions of North China, Geosci. Model Dev., 15, 8111–8134, https://doi.org/10.5194/gmd-15-8111-2022, 2022.

Gómez-Navarro, J. J., Raible, C. C., and Dierer, S.: Sensitivity of the WRF model to PBL parametrisations and nesting techniques: evaluation of wind storms over complex terrain, Geosci. Model Dev., 8, 3349–3363, https://doi.org/10.5194/gmd-8-3349-2015, 2015.

Dzebre, D. E. and Muyiwa, S. A.: A preliminary sensitivity study of Planetary Boundary Layer parameterisation schemes in the weather research and forecasting model to surface winds in coastal Ghana, Renewable Energy, 146, 66-86, 2020.

Ma, Y.-F., Wang, Y., Xian, T., Tian, G., Lu, C., Mao, X., and Wang, L.-P.: Impact of PBL schemes on multiscale WRF modeling over complex terrain, Part I: Mesoscale simulations. Atmospheric Research, 297, https://doi.org/10.1016/j.atmosres.2023.107117, 2024.

pg. 14, line 356: Quantitatively state what these deviations are instead of using qualitative language. This advice goes for the entire paper, where qualitative statements are often more common than quantitative.

Response:

Thank you for your valuable feedback. We appreciate your suggestion to incorporate more quantitative statements throughout the paper. We have thoroughly reviewed the manuscript and made the revisions.

Line 426-429: In terms of mean values, the MYJ scheme exhibits relatively smaller deviations for wind speeds below 8 m s$^{-1}$, an average deviation ranging from 0.5 to 1.25 m s$^{-1}$. In contrast, for wind speeds above 8 m s$^{-1}$, the MYNN2 scheme demonstrates the smallest deviation, with an average deviation of 2 m s$^{-1}$.

Line 384-385: primarily due to the concentration of wind speeds within the 1-4 m s$^{-1}$ range, where the cumulative frequency exceeds 0.6.

Line 394-396: The observed shape parameter is 1.79, while the shape parameters

Figure 7: There is a lot of information being shown here, which is tricky to do, but would this be better as a line plot where each line is a different PBL scheme, and the error bars are shading around those lines? That might be easier to read than ~100 bar charts.

Response:

Thank you for your suggestion to use a line plot with shading to represent the different PBL schemes. We understand that this approach could potentially make the comparison easier to read by reducing visual complexity. In fact, we tried implementing this idea (shown in Figure 2), but after careful consideration, we believe that our original method using box plots provides more detailed and comprehensive information.

In the original figure, the box plot is used to present the diurnal variation of wind speed errors corresponding to the four PBL schemes. This approach allows us to display not only the median error but also the spread of the data, including the interquartile range and outliers, which are crucial for understanding the variability and distribution of errors across different times of the day. The box plots effectively highlight the differences in performance among the PBL schemes, capturing both central tendencies and variations.

While the line plot with shading could simplify the visual presentation, it may also obscure some of the nuances in the data, particularly the distribution characteristics that are central to our analysis. Therefore, we believe that retaining the box plot format will provide a more informative and nuanced comparison of the PBL schemes, but we will add a detail description of this figure in the manuscript.

[Figure]

Figure 2. Diurnal variation of wind speed errors corresponding to four PBL schemes.

pg. 15, line 390: Perhaps the wrong word is being used here, but if the authors are going to make claims of significance, the authors should back up this statement with statistical significance tests. Otherwise, remove this statement and/or reword this sentence.

Response:

Thank you for your suggestion. we have removed such statements to avoid misinterpretation.

pg. 16, line 406: Unclassified results? What does this mean?

Response:

Thank you for pointing out the ambiguity in the term "unclassified results." What we meant by this was the simulation results obtained before applying the K-means clustering analysis. To clarify, the results from the simulations were first analyzed without any clustering (i.e., unclassified) and then compared to the results after applying the K-means clustering. In both cases, the QNSE and MYJ schemes were consistently found to be the most reliable for surface wind simulation in the Sichuan Basin.

Revised manuscript text as follows:

Line 484-486: This is consistent with the results obtained before applying K-means clustering, indicating that the QNSE and MYJ schemes are relatively stable and reliable choices for surface wind simulation in the Sichuan Basin with a model grid resolution of 0.3 km.

pg. 16, line 414: Are seasonal results not shown because there are no obvious seasonal differences?

Response:

Thank you for your comment. We have removed this statement in our manuscript.

---

## Referee Report (RR1)

**2nd Review of "Simulation performance of planetary boundary layer schemes in WRFV4.3.1 for near-surface wind over the western Sichuan Basin: a single site assessment" by Want et. al.**

**Major comments**

1. What insights does the distribution plot (Figure 5) offer? This figure is intended to evaluate the distribution of simulated wind speeds. However, it doesn't yet provide conclusive information. At first glance, results from the four PBL schemes appear generally misaligned with observations, especially regarding the $\lambda$ (scale) values. The conclusion primarily relies on the k (shape) parameter, which shows that all models appear reasonably close to the observed values. However, it remains unconvincing that QNES aligns most closely with observations. Line 495 suggests that the models tend to overestimate the occurrence of high values, underestimate low values, and match observations for values in the mid-range. Isn't this expected from the distribution plot? For instance, one can expect this conclusion from a uniform distribution.

2. Regarding the initial results (sections 3.3 and 3.4), the discussion seems to suggest that QNES outperforms the other models; however, this is not immediately evident. Aside from the distribution plot discussed above, it's unclear how the conclusion that QNES performs best at noon was reached. If my conversion is correct, at 4 UTC (corresponding to local noon), QNES does not appear to show the smallest bias.

3. I still have concern on the K-means clustering, even more confused after reading the response.

   a. In the manuscript, the authors state: *Previous studies have indicated that the simulation of meteorological elements within the boundary layer is influenced by meteorological conditions such as circulation patterns*. This statement sets the expectation that the following analysis will focus on how various meteorological conditions impact the performance of each PBL scheme. To achieve this, clustering should ideally be based on weather conditions rather than model errors, as this would more directly assess the influence of different meteorological scenarios on each scheme.

   b. The manuscript at Line 658 suggests the clustering is based on COR and RMSE, however, the response to my previous comment states the clustering is based on more other variables. If other variables are utilized, please specify. The COR and RMSE of which PBL scheme result are used?

   c. In the response, the authors state: *Additionally, it's important to note that the centroids themselves are calculated based on the mean position of all points within each cluster, and slight overlaps or close proximities between clusters can occur, especially if the clusters are not well-separated*. Based on my understanding, a k-means clustering model reaches convergence when the centroid of each cluster aligns with the center of the points assigned to it. The assignment of points to a specific centroid is determined by the minimum distance to that centroid.

   d. The discussion on three classes is actually rooted in the difference in weather conditions.

**Specific comments**

1. When discuss on the diurnal variation, please include the conversion between local time and UTC.

2. Line 562, PBL scheme?

3. Figure 9, why do the QNES results generally align with the COR gradient, except on 2022-04-14? Again, could you clarify how the k-means clustering was calculated in this context?

---

## Author Response (AR3)

**Responses to the Reviewers' Comments**

I appreciate the authors' efforts in addressing my concerns. I would like to recommend the acceptance of this manuscript for publication after addressing the minor comments below.

**Response:**

Thank you for your thoughtful review and for recognizing our efforts in addressing your previous concerns. We have carefully considered each of your comments and have made the necessary revisions to enhance the clarity and quality of our manuscript. Below, we outline our responses to your comments.

1. L140-141: Any supported references? Otherwise, the statement is opinionated.

Response:

Thank you for your feedback. In response to your question regarding supported references for the wind speed threshold of 6 m/s, we would like to clarify that there are indeed references available that outline maximum allowable wind speeds for takeoff and landing in the aircraft operating manuals provided by manufacturers. However, these wind speed limits can vary depending on the specific aircraft type.

In the context of this study, the 6 m/s threshold is specifically set for Guanghan Airport, a general aviation airport where flight training activities with small and medium-sized aircraft are the primary operations. This threshold indicates the wind speed at which adverse effects on flight activities may begin to occur. It is important to note that this figure is not a technical limit but rather a practical operational guideline designed to enhance safety during training flights. Exceeding this wind speed can impact aircraft handling and safety, particularly in a training environment where stability is crucial.

We acknowledge that our original statement lacked precision, and we have made appropriate revisions to enhance clarity. Thank you once again.

2. Table 1: "Micro-physical scheme" within the table was not changed.

**Response:**

Thank you for your comment regarding Table 1. We apologize for the oversight in my previous revision, where we mistakenly changed the incorrect location in the table.

In response to your comment, we have now corrected in Table 1.